# Oxynitrides enabled photoelectrochemical water splitting with over 3,000 hrs stable operation in practical two-electrode configuration

Yixin Xiao [1,5], Xianghua Kong[2,5], Srinivas Vanka[1,5], Wan Jae Dong [1,5], Guosong Zeng[3], Zhengwei Ye[1], Kai Sun[4], Ishtiaque Ahmed Navid[1], Baowen Zhou [1], Francesca M. Toma [3], Hong Guo[2] ✉ & Zetian Mi [1] ✉

Solar photoelectrochemical reactions have been considered one of the most promising paths for sustainable energy production. To date, however, there has been no demonstration of semiconductor photoelectrodes with long-term stable operation in a two-electrode configuration, which is required for any practical application. Herein, we demonstrate the stable operation of a photocathode comprising Si and GaN, the two most produced semiconductors in the world, for 3,000 hrs without any performance degradation in two-electrode configurations. Measurements in both three- and two-electrode configurations suggest that surfaces of the GaN nanowires on Si photocathode transform in situ into Ga-O-N that drastically enhances hydrogen evolution and remains stable for 3,000 hrs. First principles calculations further revealed that the in-situ Ga-O-N species exhibit atomic-scale surface metallization. This study overcomes the conventional dilemma between efficiency and stability imposed by extrinsic cocatalysts, offering a path for practical application of photoelectrochemical devices and systems for clean energy.

The photoelectrochemical (PEC) pathway for the generation of clean chemicals and fuels, e.g., $H_2$ production from solar water splitting and liquid fuel generation from $CO_2$ reduction, has attracted tremendous attention over the past decades[1–4]. Although significant progress had been made in improving the solar-to-hydrogen (STH) efficiency of PEC devices[5–7], the stability of these devices has remained a daunting challenge, preventing any practical, large-scale deployment of this promising technology[8–11]. As a diagnostic tool, three-electrode configuration has been commonly employed to study the stability of semiconductor photoelectrodes. The incorporation of protection schemes[12] has significantly improved the stability of various

materials[6,13,14] in three-electrode PEC testing configurations[9,12,15–18]. Three-electrode PEC testing, however, does not consider the performance of the counter electrode[2,19]. Nor does it describe the overall PEC system stability[2,19]. Moreover, since the protection layers are not catalytically active for hydrogen evolution reaction (HER), additional extrinsic catalysts have been required on the photoelectrodes. Shown in Supp. Info. Table S1, previously reported high-efficiency photoelectrodes that have demonstrated reasonable stability in a three-electrode configuration exhibit a rather poor stability performance when measured in a practical system configuration, i.e., a two-electrode configuration[5–7,20,21], due to factors such as the charge

[1]Department of Electrical Engineering and Computer Science, University of Michigan, Ann Arbor, 1301 Beal Avenue, Ann Arbor, MI 48109, USA. [2]Department of Physics, McGill University, 3600 University Street, Montreal, Quebec H3A 2T8, Canada. [3]Lawrence Berkeley National Laboratory, Chemical Sciences Division, 1 Cyclotron Road, Berkeley, CA 94720, USA. [4]Department of Materials Science and Engineering, University of Michigan, 2300 Hayward Street, Ann Arbor, MI 48109, USA. [5]These authors contributed equally: Yixin Xiao, Xianghua Kong, Srinivas Vanka, Wan Jae Dong. ✉e-mail: hong.guo@mcgill.ca; ztmi@umich.edu

build-up within the cell, solution resistance, and inherent chemical instability of the photoelectrode. These studies further suggest that the three-electrode measurement is neither sufficient for nor relevant to the overall stability of the eventual commercial deployment of PEC water-splitting systems. In contrast, testing under the two-electrode configuration gives the actual efficiency and durability of the entire PEC cell[19]. To date, however, there has been no demonstration of semiconductor photoelectrodes, or suitable protection schemes, that can enable long-term stable and efficient operation in a practical two-electrode configuration.

Previous studies have been largely focused on metal-oxide, Si, and III-V-based semiconductor photoelectrodes. Recently, a new class of semiconductor photoelectrodes, consisting of metal-nitride nanostructures, has drawn considerable attention. Metal-nitrides, especially III-nitrides, e.g., InGaN, have tunable energy bandgap across the entire solar spectrum[22,23]. Moreover, GaN and Si, the two most produced semiconductors in the world, can be seamlessly integrated to achieve high-efficiency solar water splitting with proven manufacturability, scalability, and relatively low cost[24–30]. Zeng et al. showed that GaN possesses a unique *self-improving* property, i.e., the PEC performance showed an improvement, instead of degradation, over the course of a three-electrode chronoamperometry (CA)[31]. This quite unusual behavior was attributed to the formation of oxynitride on the nonpolar and semipolar GaN surfaces during PEC reactions. At the same time, the measurements were performed in a three-electrode configuration where the morphology of GaN was quasi-film, for which a relatively small fraction of the surface in contact with the electrolyte was the active nonpolar surfaces. It has remained unknown whether such a unique *self-improving* behavior can be maintained or even enhanced and long-term stability can be achieved for GaN morphologies in which the active nonpolar surfaces dominate under practical two-electrode conditions. Moreover, the underlying mechanism for the oxynitride formation, its atomic origin and catalytic properties, and its dependence on surface polarity and configuration have remained unknown.

Herein, we have performed a detailed study of the GaN nanowires (NWs) array grown on Si photocathode for self-improving behavior and long-term stability. The GaN NW/Si photocathode where nonpolar sidewalls dominate shows photoelectrochemical characteristics that are dramatically improved compared to the GaN film on Si photocathode studied before. The experiments in three-electrode configurations confirm that the GaN/electrolyte interface is necessary for the self-improvement effect, the speed of which scales super-linearly with increases in photocurrent density achieved via concentrated sunlight illumination. X-ray photoelectron spectroscopy (XPS) measurements further confirm that, during the initial hours of the stability testing, there is an in-situ formation of new gallium oxynitride species on the nonpolar *m*-plane of GaN nanowires, which leads to improved *J-V* characteristics, including a higher photocurrent density and more positive onset potential. We have further demonstrated stable operation for 3000 h without any performance degradation in practical two-electrode configurations, exceeding the previously best-reported stability of 300 h in a two-electrode configuration by an order of magnitude[32]. Significantly, the measurements were performed without the incorporation of any metal catalyst protection, revealing the intrinsic stability of GaN/Si photoelectrodes. First principles density functional theory (DFT) studies further revealed the formation mechanism, atomic origin, electronic structure, and catalytic properties of the unique GaON species. We show that the in-situ formation of atomic-scale GaON nanoclusters on N-terminated GaN nanowires takes place when O atoms partially replace the N atoms on the non-polar GaN *m*-plane. The incorporation of O atoms on GaN not only reduced surface band bending, but also created atomic-scale localized nanoclusters of semiconductor surface metallization, i.e., GaON species, which naturally act as reduction reaction sites. This study has overcome the stability bottleneck of semiconductor photoelectrodes,

offering a path for practical application of photoelectrochemical devices and systems for clean energy.

## Results
### GaN/Si photocathode: three-electrode characterization

The synthesis and fabrication of $n^+$-GaN nanowires/Si $p$-$n$ junction photocathodes are discussed in the Methods section. Scanning electron microscope (SEM) image of the as-grown GaN nanowires is shown in Fig. S1. The nanowires have an average length of ~600 nm and diameter of ~100 nm. The photocathode was first evaluated in a three-electrode configuration with an iridium oxide ($IrO_x$) counter electrode and a Ag/AgCl reference electrode under AM 1.5 G 1-sun illumination at an angle perpendicular to the photocathode wafer (Fig. 1a). In this device, incident solar photons are absorbed by the Si wafer, and photo-generated electrons are extracted by $n^+$-GaN nanowires to drive proton reduction[12,33]. Chronoamperometry (CA) curve of the photocathode at −0.4 V *vs.* reversible hydrogen electrode ($V_{RHE}$) shows a rapid increase in photocurrent density from 0.6 to ~35 mA/cm$^2$ over 40–50 h (Fig. S2). Linear sweep voltammetry (LSV) curves of the photocathode show in-situ gradual improvement of onset potential ($V_{on}$, voltage at 1 mA/cm$^2$) and photocurrent density (Fig. 1b). After 100 h, the measured $V_{on}$ = 0.36 $V_{RHE}$ and photocurrent density = ~30 mA/cm$^2$ at 0 $V_{RHE}$ are among the best performances of Si photocathode without the incorporation of extrinsic catalysts and are within the performance range of Si photocathodes with noble metal catalysts (Figure S3 and Table S2). This is a significant advancement from the previous work on quasi-film GaN/Si photocathode due to the unique morphology and polarity of vertically aligned GaN nanowires for which the active m-plane sidewalls (to be discussed in Theoretical calculations) dominate and which enhance mass transport due to increased porosity of the NW array compared to the quasi-film.

In the Nyquist plots (Fig. S4), the radius of the semicircle gradually decreased with the increase in reaction time, indicating a decrease in charge transfer resistance and an accelerated electron transfer from the photocathode to the protons. Faradaic efficiency of $H_2$ increased from 52% to >95% during the initial 1.5 h of reaction at −0.4 $V_{RHE}$ and stabilized thereafter (Fig. S5). We also found that the performance was not improved at open-circuit potential even under light illumination, demonstrating that the photocurrent plays a key role in enhancing the catalytic activity of GaN NW/Si photocathode (Fig. S6). In order to confirm experimentally whether the self-improvement originated from in-situ modification of GaN surface, we coated a thin (2 nm) passivation layer of $Al_2O_3$ on the GaN NWs by atomic layer deposition and carried out PEC $H_2$ evolution reaction at −0.4 $V_{RHE}$ under one-sun illumination (Fig. S7). Interestingly, there was negligible improvement of photocurrent density and $V_{on}$ even after 24 h of reaction. In contrast, after Pt cocatalyst deposition, the Pt/$Al_2O_3$/GaN NW/Si photocathode showed a LSV curve like that of Pt/GaN NW/Si. As such, the $Al_2O_3$ passivation layer allows for efficient transfer of the photogenerated electrons from the GaN NWs to catalytic sites but effectively prevents the in-situ surface modification of GaN. We have further confirmed via XPS and scanning transmission electron microscopy (STEM) energy dispersive X-ray spectroscopy (EDS) that there was no Ir contamination on the GaN NW from the counter electrode ($IrO_x$) after 48 h of chronoamperometry (Fig. S8).

From these results, it is identified that there are two necessary conditions for the self-improvement of GaN NW/Si photocathode: (1) photocurrent and (2) exposure of the GaN surface to the electrolyte. With one-sun light intensity, a relatively prolonged duration (>10 h) was needed to obtain high $H_2$ evolution activity (i.e., $V_{on}$ > 0 $V_{RHE}$) possibly because the rate of surface modification was limited by the small photocurrent. Boosting the photocurrent density by irradiating with the concentrated solar light can shorten the time duration to achieve the saturation in self-improvement. Hence, we designed a flow cell which can afford rapid replacement of reactants and products at the

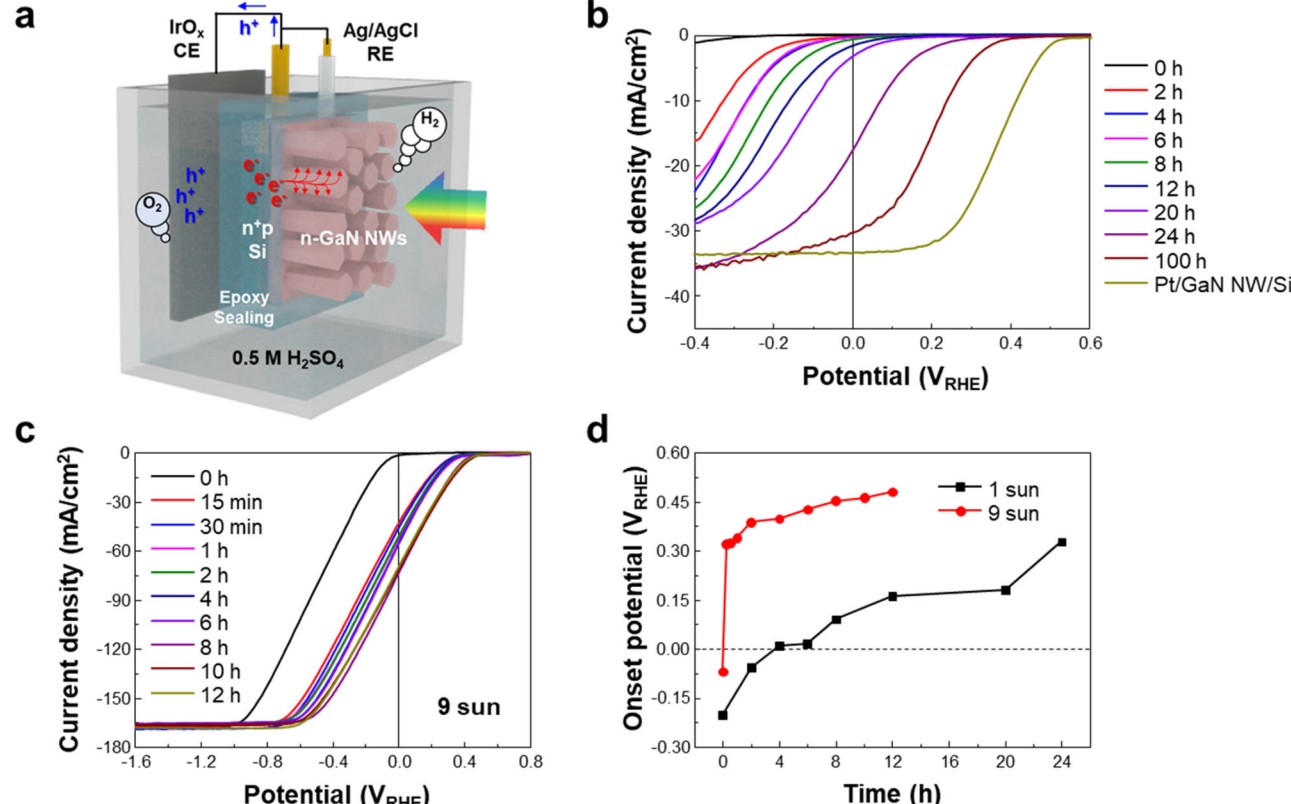

**Fig. 1 | Photocathodes evaluated in a three-electrode configuration.**
**a** Schematic illustration of photoelectrochemical $H_2$ evolution reaction in a three-electrode configuration. GaN NW/Si photocathode, iridium oxide ($IrO_x$), and Ag/AgCl were used as the working, counter, and reference electrode, respectively. **b** Linear sweep voltammetry (LSV) curves of the GaN NW/Si photocathode after

reaction for 0 – 100 h and Pt-loaded GaN NW/Si photocathode. **c** LSV curves of the GaN/Si photocathode measured under 9 sun (900 mW/cm²) solar light in a flow cell. **d** Plot of onset potential *vs.* reaction time under 1 (black curve) and 9 sun (red curve) illumination.

electrode (Fig. S9a), and measured LSV curves at intervals of reaction time under nine-sun solar light (Fig. 1c). Interestingly, $V_{on}$ > 0.3 $V_{RHE}$ was achieved within 15 min of reaction. The plot of $V_{on}$ vs. reaction time shows that there was a super-linear correlation between the $V_{on}$ improvement speed and light intensity, thereby resulting in ~100 times faster GaN surface modification under nine-sun illumination compared to one-sun illumination (Fig. 1d). The high photocurrent density (~165 mA/cm² at −0.3 $V_{RHE}$) obtained by the concentrated solar light not only boosted the speed of the photocathode self-improvement but also increased the $H_2$ production rate by ~9 times (Fig. S9b). Moreover, the GaN NW/Si operated stably more than 500 h with very high photocurrent density (> 150 mA/cm²) at −0.4 $V_{RHE}$ under the concentrated solar light (Figs. S9c and S9d). The produced total $H_2$ gas is similar to the amount of $H_2$ produced for 4,500 h under one-sun illumination.

## GaN/Si photocathode: two-electrode characterization
The photocathode was further tested in two-electrode configurations under AM 1.5 G one-sun illumination. Figure 2a shows the chronoamperometry of the photocathode for the first 10 h at −2.3 V *vs.* $IrO_x$ under AM 1.5 G one-sun illumination in 0.5 M $H_2SO_4$ with 0.2 mM Triton X-100 as surfactant (see Methods). The rather large bias was chosen to ensure operation at the saturation current density for the majority of the ensuing 3000 h long-term stability testing (discussed in detail next section); we have confirmed that the photocathode produces hydrogen as efficiently at lower biases as well (Fig. S10). It is noted that, similar to the photocathode under three-electrode configurations, the photocurrent density also increased considerably over this initial 10 h of the CA stability test. Figure 2b shows the Faradaic efficiency and $H_2$ evolution for the sample between 0 h and 10 h of

chronoamperometry at −2.3 V *vs.* $IrO_x$ in 0.5 M $H_2SO_4$ under AM 1.5 G one sun illumination. The Faradaic efficiency for this duration was 89–100%. Due to the formation of oxynitride species, Faradaic efficiency was initially lower than 100%, but it increased steadily and reached 100% by the end of 10 h of chronoamperometry.

Similar to the results in three-electrode configurations, the LSV measurements (Fig. 2c) under AM1.5 G one-sun illumination at 0 h (red curve) and 10 h (blue curve) clearly show an improvement in fill factor, a positive shift of ~0.5 V in $V_{on}$, and an increase in photocurrent density. Furthermore, electrochemical impedance spectroscopy measurements, in the form of Nyquist plots (Fig. 2d) at 0 h (red curve) and 10 h (blue curve)[33], show a drastic reduction in charge transfer resistance by nearly two orders of magnitude, for the 10 h tested sample compared to the measurements taken at 0 h. These results point to the possibility of the formation of new species on the surfaces of GaN nanowires similar to that responsible for the self-improvement observed under a three-electrode configuration[31].

The XPS measurements, shown in Fig. 3, are taken on GaN nanowires/Si photocathode samples before and after 10 h of stability test. Figure 3a and b show the XPS *O 1 s* peaks taken at an incident angle of 60°, where *m*-plane surfaces of nanowires were predominantly measured. Apart from the O-Ga (red curve)[34,35] and O-H (blue curve) peaks, there is an additional deconvoluted O-Ga-N peak[36] (at -531.6 eV) for the 10 h tested sample (Fig. 3b), compared to the pristine sample (Fig. 3a). The O-Ga-N peak shows that a new oxynitride species formed along the nonplanar (*m*-plane) surface of GaN nanowires during the 10 h stability experiment. It is to be noted that the deconvoluted Ga-O peak (for the pristine sample) is due to the exposure of as-grown GaN surface to ambient conditions before transferring to XPS chamber[37]. As shown in

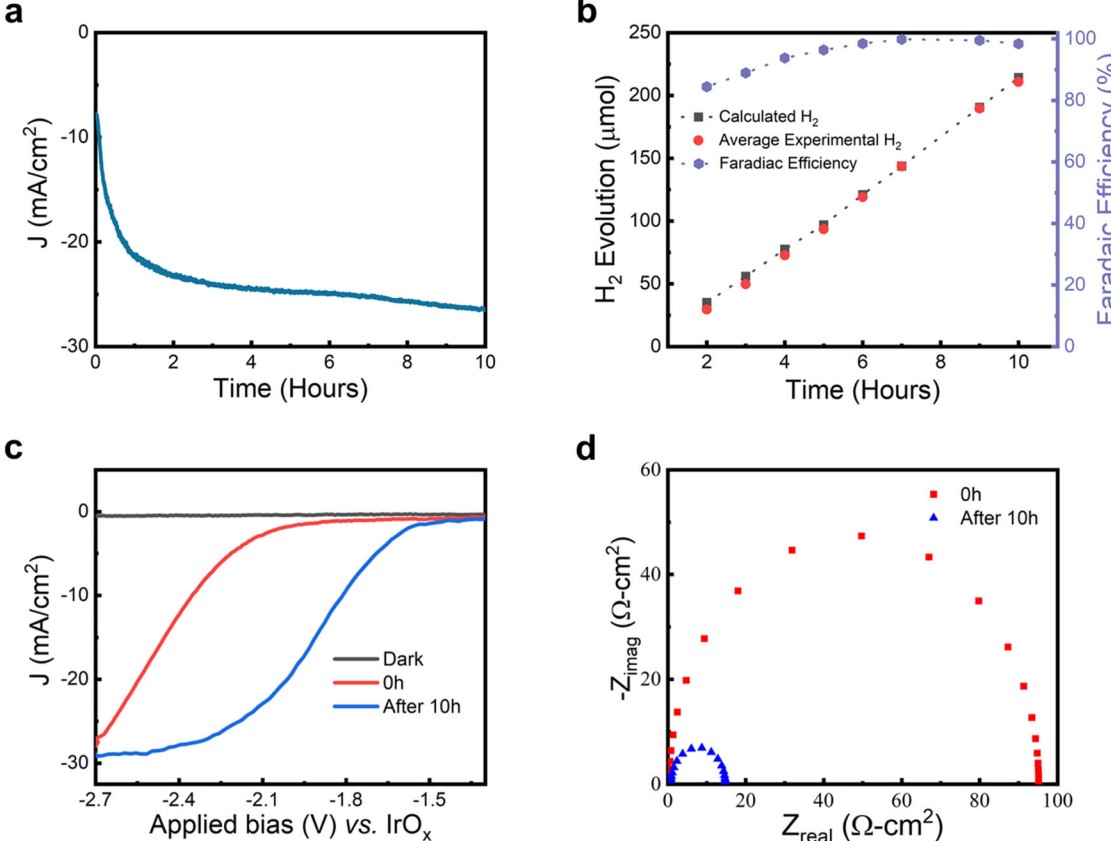

**Fig. 2 | Photocathodes evaluated in a two-electrode configuration. a** The initial 10 h of the chronoamperometry (CA) stability test of the photocathode at −2.3 V *vs.* IrO$_x$ under AM 1.5 G one-sun illumination in 0.5 M H$_2$SO$_4$ with 0.2 mM Triton X-100. **b** Faradaic efficiency measurements of the first 10 h of the CA. The red dots represent the average amount of H$_2$ measured at different times via gas chromatography, and the black dotted curve is the amount of H$_2$ calculated from the photocurrent over time. **c** LSV curves of the photocathode before (red curve) and after 10 hours (blue curve) of CA under AM 1.5 G one-sun illumination and dark (black curve) conditions. **d** Nyquist plots of the photocathode before (red curve) and after 10 h (blue curve) of CA under AM 1.5 G one-sun illumination.

Fig. 3b, the relative O-Ga peak intensity has considerably reduced for the 10 h tested sample compared to the pristine sample, likely due to the dissolution of oxide in acidic conditions[38] and conversion of some oxide into oxynitride species. Similar conclusions can also be drawn from the *Ga 3d* spectra given in Supp. Info. Figure S11. As shown in Figure S11a, for the pristine sample, the two deconvoluted peaks correspond to Ga-N (cyan curve) and Ga-O (magenta curve) at 20.2 eV and 21 eV, respectively[37,39]. Figure S11b shows that, like the *O 1 s* spectra, an additional deconvoluted Ga-N-O peak at 20.8 eV between the Ga-N and Ga-O peaks emerges for the *Ga 3d* spectrum of the 10 h tested sample.

Figure 3c shows the valence spectra for both the pristine and the tested sample. In Fig. 3c, the valence spectrum measurement for the tested sample showed an increase of ~ 0.5 V in $E_{FS} - E_{VS}$ compared to the pristine sample, where $E_{FS}$ is the surface Fermi level and $E_{VS}$ is the surface valence band maximum. This valence spectrum shift is consistent with the V$_{on}$ shift, shown in Fig. 2c. Theoretical calculations, to be discussed below in greater detail, point to a reduction of the conduction band barrier height due to oxynitride formation that can contribute to the increase in the $E_{FS} - E_{VS}$ value after 10 h of chronoamperometry. The conduction band barrier reduction helps improve the charge transfer kinetics for H$_2$ production, which is reflected in the LSV curves shown in Figs. 1b and 2c.

**Long-term stability testing**
The chronoamperometry (CA) stability measurements were taken at a constant applied potential of −2.3 V *vs.* IrO$_x$ under AM1.5 G one-sun illumination in 0.5 M H$_2$SO$_4$ with 0.2 mM Triton X-100. The electrolyte solution is replaced afresh after every 20–24 h of CA[12]. Figure 4a shows

the photocurrent density throughout 3000 h for the GaN NW/Si photocathode. A summary of the measured V$_{on}$ and $J$ at −2.3 V *vs.* IrO$_x$ at the end of integer multiples of 100 h throughout the 3000 h chronoamperometry is given in Supp. Info. Table S3 and plotted in Fig. 4a. At the beginning of the stability experiment (0$^{th}$ hour), the V$_{on}$ was ~ −1.8 V *vs.* IrO$_x$ and the photocurrent density of the sample was ~8 mA/cm$^2$ at −2.3 V *vs.* IrO$_x$. The *J-V* characteristics (Fig. 4b) for the photocathode after 10 h (blue curve) of chronoamperometry clearly show a dramatic improvement over the 0$^{th}$ hour *J-V* curve (red curve). As discussed earlier, this improvement is attributed to the formation of oxynitride species on the *m*-plane of GaN nanowires. Further continuation of the CA stability experiments under the same experimental conditions shows that the photocurrent density reached its saturation value of ~30 mA/cm$^2$ after ~ 40 h (Figure S12). It is to be noted that the *J-V* characteristics after 1000 h (purple curve) and 3000 h (cyan curve in Fig. 4b) show near-identical V$_{on}$ ~ −1.35 V *vs.* IrO$_x$ and fill factor compared to the 10 h (blue curve). It can be thus concluded that, once formed on GaN nanowires *m*-plane surfaces, the GaON species are robust against both the continuation of stability experiments and the exposure to air during the routine electrolyte change every 20–24 h of CA. During the entirety of the 3000 h chronoamperometry (see Fig. 4a), the photocurrent density varied by ±10% of the average value (~29 mA/cm$^2$). The exact duration at which catastrophic device failure may occur for this photocathode is yet to be determined. Future work will focus on using harsher conditions like high temperature and concentrated sunlight[10,40] to better quantify the durability of this device.

To the best of our knowledge, the ultrahigh stability of ~3000 h for Ga(O)N nanowires/Si is the greatest stability duration measured for

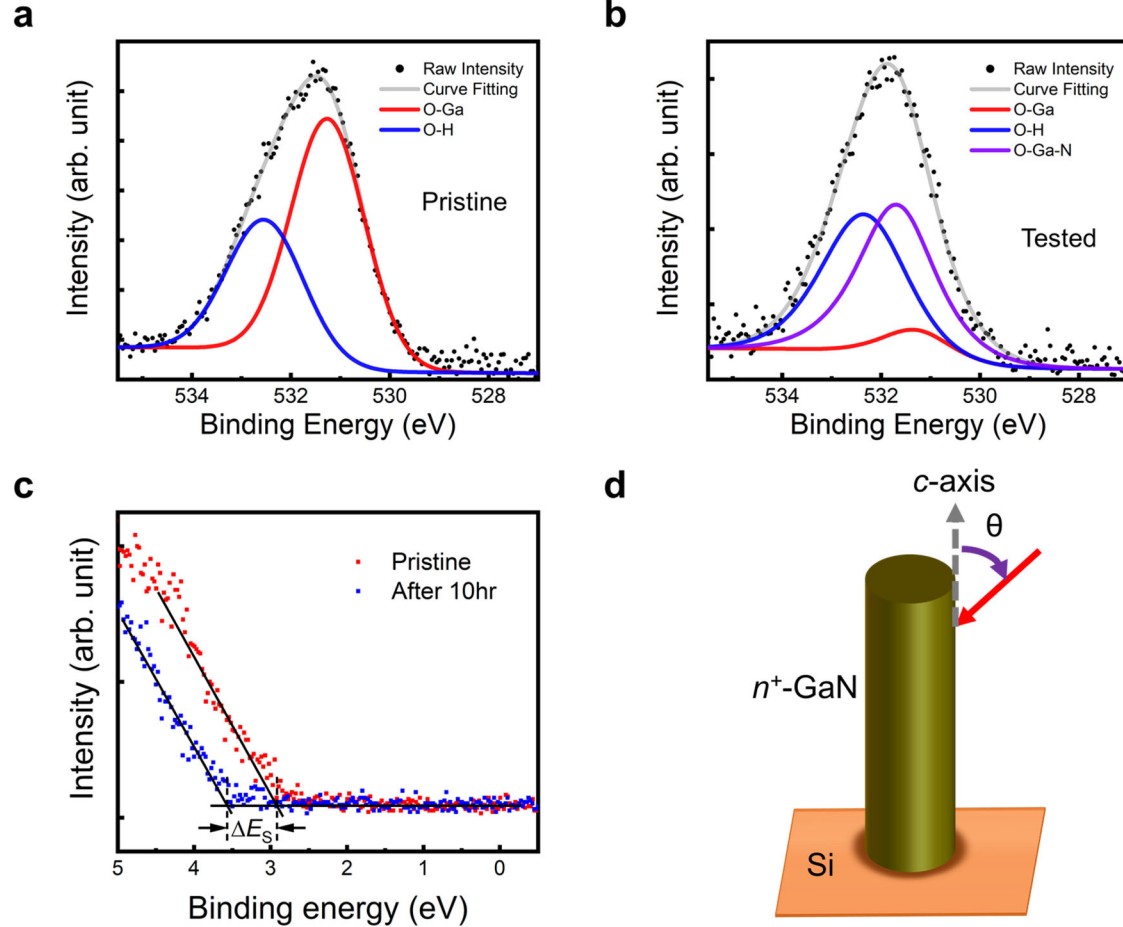

**Fig. 3 | X-ray photoelectron spectroscopy of the photocathode before and after stability measurements. a, b** XPS *O 1s* spectra of the $n^+$-GaN nanowires/Si photocathode as-grown (**a**) and tested (**b**) under conditions described in the caption of Fig. 2, taken with the incident angle $\theta = 60°$. The definition of the angle $\theta$ is schematically shown in (**d**). The as-grown photoelectrode (**a**) showed deconvoluted peaks at 531.3 eV and 532.6 eV for O-Ga (red curve) and O-H (blue curve), respectively. After 10 hours of chronoamperometry (CA) (**b**), an additional deconvoluted peak at 531.6 eV corresponding to gallium oxynitride species (violet curve)

emerged. The grey curves in (**a**) and (**b**) are the fitted curves of their respective raw data. **c** Valence band maximum measurements of the photocathode before (red dots) and after 10 hours of CA (blue dots), where the quantity $E_{FS} - E_{VS}$ increased by $\triangle E_S \approx 0.5$ eV. The intersection between the background intensity flatline (at binding energy less than 2 eV) and the linear fit of the onset of the photoelectron signal intensity is the position of the surface Fermi level ($E_{FS}$) relative to the surface valence band maximum ($E_{VS}$).

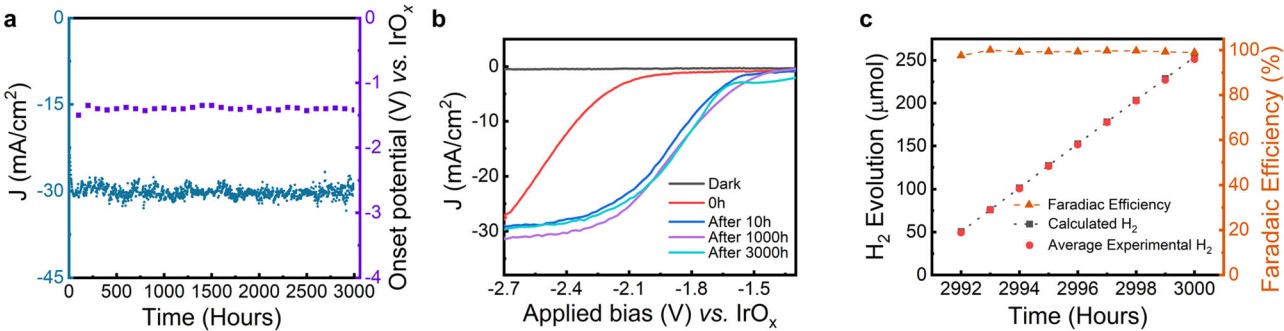

**Fig. 4 | Two-electrode ultralong stability experiments of the $n^+$-GaN nanowires/ Si photocathode. a** Chronoamperometry (CA) shows 3000 h stability for the $n^+$-GaN nanowires/Si photocathode at −2.3 V *vs.* IrO$_x$ under AM 1.5 G one-sun illumination in 0.5 M H$_2$SO$_4$ with 0.2 mM Triton X-100. The material stability is further highlighted by the stability of the onset potential over the course of the CA. **b** Linear scan voltammograms of the photocathode at 0 h (red curve), after 10 h (blue), after

1000 h (purple curve), and after 3000 h (cyan curve) under AM 1.5 G one-sun illumination and dark (black curve) conditions. **c** Faradaic efficiency measurements of the last 10 h of the 3000 h CA. The red dots represent the average amount of H$_2$ measured at different times via gas chromatography, and the black dotted line is the amount of H$_2$ calculated from the photocurrent over time.

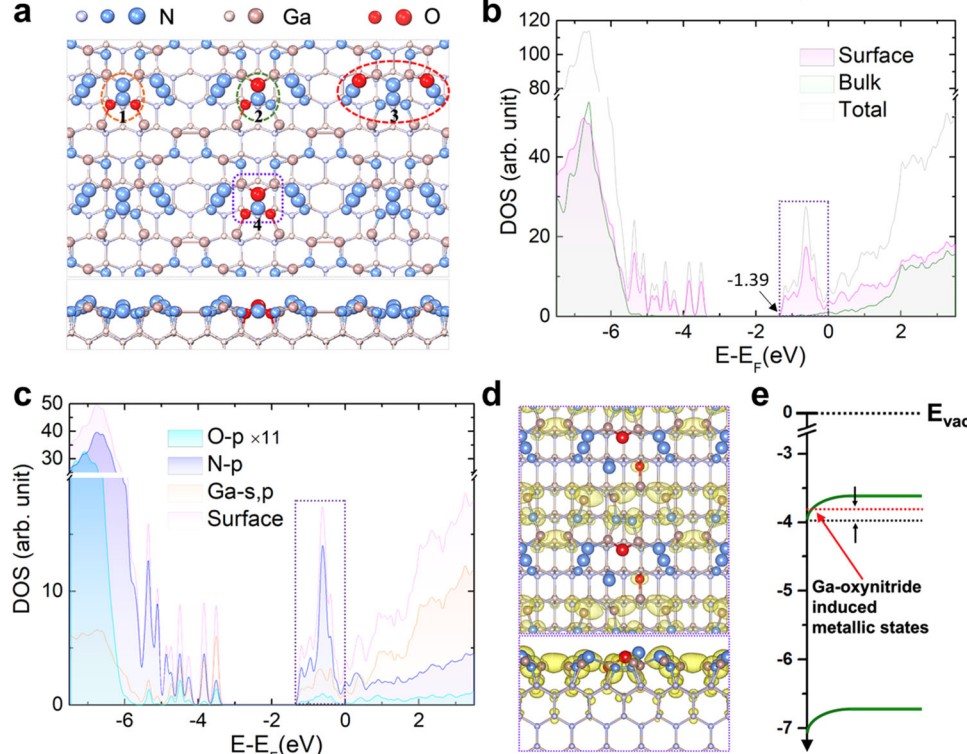

**Fig. 5 | Calculated electronic structures of metal oxynitride species featured on GaN *m*-plane. a** Top (upper panel) and side (lower panel) view of proposed configurations of GaN *m*-plane with surface metal oxynitride nanoclusters. **b** The density of states (DOS) of metal oxynitride species featured GaN *m*-plane. The contributions from the surface and bulk atoms are marked with pink and green curves, respectively. **c** Orbital-projected DOS of Ga-oxynitride. **d** Spatial structure of charge densities of Ga-oxynitride in the energy range of −1.39 eV -0 eV as marked with purple dashed rectangles in (**b, c**). The top and bottom panels are respectively the top and side view of the spatial structure. The value of the isosurfaces is 0.001 eÅ⁻³. **e** Band diagram of GaN with Ga-oxynitride-induced metallic states.

any photoelectrode in a two-electrode configuration at any bias under AM 1.5 G one-sun illumination in any electrolyte solution (see Supp. Info. Tables S1 and S4)[1,7,41]. This extraordinary stability of the GaN NW/ Si photocathode can be attributed to material properties of the GaN nanowires, such as strong ionic bonds, free of dislocations, and a unique N-termination on both the *c*-plane and the *m*-plane[12,25,33,38]. By comparing the *J-V* characteristics of the photocathode after 10 h (blue curve in Fig. 4b) and 3000 h (cyan curve in Fig. 4b) of CA, it can be concluded that the LSV curves remained almost identical 10 h−3000 h due to the excellent stability of oxynitride species on the GaN nanowires *m*-plane. In addition, Faradaic efficiency for hydrogen evolution during the last 10 h of the 3000 h chronoamperometry (Fig. 4c) remained constant, which, together with the constant photocurrent, confirms the photocathode's stability for the entirety of the 3000 h CA. STEM measurements (Fig. S13a) indicate that the nanowire dimensions remained unchanged with length ∼600 nm and diameter ∼100 nm after 3000 h of chronoamperometry. In addition, the SEM image of the 3000 h tested photoelectrode (Fig. S13b) shows virtually no change in either GaN nanowires coverage on Si or nanowire morphology. Dissolved Ga in the electrolyte was analyzed using inductively coupled plasma mass spectroscopy (ICP-MS) at different stages of the entire CA stability experiments. Supp. Info. Table S5 shows negligible amount of the dissolved Ga concentrations of 10 − 13 nmol for different runs. These results show that the nanowires have remained stable throughout the CA experiments, consistent with the structural characterizations.

## Theoretical calculations

To gain an atomic-level understanding of metal oxynitride species on the GaN nanowires, we performed density functional theory (DFT) calculations (see Methods section) on the formation of oxynitride species and their electronic and catalytic properties associated with water splitting. We considered four possible atomic configurations of GaON-featured GaN *m*-plane (marked with orange, green, red, and purple circles in the upper panel of Fig. 5a respectively) based on previously reported N-rich GaN surfaces[31,38]. Previous studies have suggested that the *m*-plane surfaces of GaN nanowires grown via plasma-assisted molecular beam epitaxy (PAMBE) under nitrogen-rich conditions are characterized by the presence of nitrogen nanoclusters. This exotic surface feature of the GaN nanowires facilitates further incorporation of oxygen at the nanowire *m*-plane sidewalls. These nitrogen clusters are relatively isolated on the surface, which are spatially conducive to achieving the replacement of the N atom with an O atom to form Ga-O-N species. The formation energy for the four proposed GaON configurations in Fig. 5a was calculated to be −4.07 eV, −4.29 eV, −3.95 eV, and −2.66 eV. The negative formation energy indicates that the introduction of the O atom to the N-rich GaN surface is a thermodynamically favored process, which also agrees with the consensus that metal-O bonds are stronger than metal−N bonds[42]. The stability of Ga-O-N species was therefore verified. We further calculated the density of states of N-rich GaN *m*-plane before (Fig. S14) and after oxidation (Fig. 5b and S15). A fascinating characteristic was found that the Fermi level of GaON shifted upward into the conduction band (with a value of 1.39 eV for the second configuration), rendering the oxygen-incorporated *m*-plane surface metallized. This metallic nature could be simply understood to come from an effective *n*-type doping where the O atom owns one more electron than the N atom it replaces. Since the valence band of GaN has been fully occupied before oxidation, this extra electron brought by the introduction of the O element could be filled only in the conduction band. Since that the second configuration (marked with the green circle in Fig. 5a) was the most negative in formation energy and therefore the most

thermodynamically favored among the four proposed configurations, further calculations were based on the second configuration (Supp. Info. Table S6). From the calculated orbital-projected density of states (OPDOS) and atom-projected density of states (APDOS), we found that only the *p* orbital of O, N atoms, and *p* and *s* orbitals of Ga atoms in the surface region are responsible for the metallic manifestation, which indicates a downward band bending (Fig. 5b–e) and thus benefits the reduction reaction. Meanwhile, the bandgap of GaN in the outermost surface of the nanowires is effectively narrowed from 3.19 eV (Fig. S16) to 1.83 eV due to the emerging metallic states brought by Ga oxynitride species formation. For a more intuitive view of the metallic surface, we plotted the real-space-distributed charge densities of the conduction band around Fermi level ($-1.39$ eV $< E\text{-}E_F < 0$ eV, marked by the purple dashed rectangle in Fig. 5b, c) in Fig. 5d. The metallic property directly corresponds to the surface GaON species, which naturally act as an electron sink and serve as atomic-scale reduction reaction sites. Therefore, through first-principles calculations, we determined that the surface configuration of the N-rich GaN *m*-plane featured with GaON nanoclusters should largely enhance the stability and efficiency of the solar-powered artificial photosynthesis.

## Discussion

Finally, we discuss the unique advantages of in-situ formation of GaON nanoclusters on N-terminated GaN nanostructures, compared to conventional nitridation schemes of III-V photoelectrodes and $Ga_2O_3$ powders to improve the stability of photoelectrochemical reactions. It is well established that nitrogen-containing photocatalysts have stable and efficient operation in harsh solar water splitting conditions compared with traditional metal oxides and III-V compounds[30,38]. In addition, it has been demonstrated that the incorporation of nitrogen species improves stability and is essential for efficient light absorption by narrowing the bandgap[30,43]. Previous studies explored the metal-organic chemical vapor deposition (MOCVD) growth of GaPN epilayers, with 0.2%–2% nitrogen, on GaP substrates for protecting the III-V photoelectrodes[43]. While nitrogen incorporation at these low levels improved the stability of the material against photocorrosion, further nitrogen incorporation in these structures created a huge lattice mismatch leading to surface defects and increased photocorrosion[43]. To date, it has remained elusive to achieve long-term stability not only for nitridated III-V photoelectrodes, but for high-efficiency photoelectrodes in general. To enhance the stability of the photoelectrode, much attention has been given to the employment of oxide layers, via methods such as atomic layer deposition of $TiO_2$ and $Al_2O_3$[13,15,17,44], for protection against various corrosion pathways, such as photo-oxidation and reaction with electrolytes[45,46]. The employment of such a foreign protection layer (on the photoelectrode) often faces the undesirable tradeoff between preservation of hard-won photoelectrode efficiency and realistic enhancement of photoelectrode stability, since such oxide layers are by design chemically inactive on the surface and very often poor in electrical conductivity as well. As demonstrated in this work through the exemplary atomically thin, catalytically active GaON species on the sidewalls of the GaN NW/Si photoelectrode in practical two-electrode configurations, the in-situ formation of native surface catalysts provides a compelling answer to the dilemma. Prior to this work, GaON has been predominantly fabricated by nitridation of $Ga_2O_3$ that requires annealing at high temperatures for prolonged durations[36,47,48]. This fabrication route is energy-intensive and results in low yield and quality, leading to inferior photocatalyst performance and lower stability (Supp. Info. Table S4). The N-terminated GaN nanowires on Si presented in this work have unique advantages of N-rich *m*-plane sidewalls[38], strong ionic bonds[49], nearly perfect band alignment[33] and defect-free single-crystal wurtzite structure[38]. These GaON nanocluster species act as catalysts to improve the charge carrier kinetics and operate efficiently for thousands of hours without the need for additional catalyst regeneration. As such, GaON nanoclusters on

N-terminated GaN nanowires are an excellent platform for providing ultrahigh stability and efficient surface charge transfer kinetics under practical 2-electrode PEC conditions.

In conclusion, we have demonstrated ultra-stable in-situ self-improvement of GaN NW/Si photocathode for PEC $H_2$ evolution reaction. We have identified morphology and light intensity as factors for enhancing or expediting the self-improvement effect. Owing to the unique physical and chemical nature of GaN nanowires, a great advancement has been established compared to the previous Si photocathodes. We also have shown that the GaN nanowires/Si photocathode, without any foreign co-catalysts, can achieve unprecedentedly ultrahigh long-term stability of 3000 h in practical two-electrode conditions under AM 1.5 G one-sun illumination with photocurrent densities $>25$ mA/cm$^2$ and a Faradaic efficiency of ~100%. During the stability experiments, the photocathode exhibited a self-improvement mechanism in the formation of new oxynitride species on the *m*-plane of GaN nanowires. Through DFT calculations, we discovered that the formation of Ga-O-N species on the N-terminated GaN *m*-plane provided natural atomic-scale reduction reaction sites since the emerging oxynitride species exhibit metallic properties. Even better, these localized metallic surface states cause downward band bending, which further facilies the reduction reactions. In future work, detailed studies are required to understand the true durability of this new class of photoelectrodes.

## Methods

### Si solar cell fabrication

Double-side polished 4" *p*-type Si (100) wafers (University Wafers, thickness: 254–304 μm; resistivity: 1 – 10 Ω•cm) were first RCA cleaned and then loaded to a CMOS grade oxidation furnace to form 250 to 300 nm thick $SiO_x$ at 1100 °C. The oxide grew on both sides of the double-side polished wafers. Subsequent lithography and wet etching steps led to only one side of the wafers having $SiO_x$, (with the other side being exposed *p*-Si). After another RCA cleaning, the wafers were loaded in a CMOS grade phosphorus diffusion furnace (Tempress diffusion furnace) to form $n^+$-Si doping at 950 °C for 20 min. These wafers were then cleaned in buffered HF solution for 5–10 min to remove the $SiO_x$ and residual silicate built on the $n^+$-Si side. Four-point probe and secondary-ion mass spectrometry (SIMS) measurements on the $n^+$-Si side of the cleaned $n^+\text{-}p$ Si wafers showed sheet resistance values of 8 –11 Ω/sq with a thickness of ~0.6 μm and donor concentration of ~$1 \times 10^{20}$ cm$^{-3}$. The 4" wafer was then diced into quarter wafers which were subsequently cleaned following standard solvent/acid protocols before being loaded into the MBE chamber[33].

### Growth of $n^+$-GaN nanowires on Si

Before beginning the growth of $n^+$-GaN nanowires on Si substrates, we first grew an N-terminated thin GaN quasi-film on Si for 15 min with a substrate temperature at ~735 °C, Ga beam equivalent pressure (BEP) of ~$2.2 \times 10^{-7}$ torr, Si cell (*n*-type dopant) temperature at 1250 °C, and a nirtrogen flow rate of 0.45 standard cubic centimeter per minute (sccm). The incorporation of a thin GaN quasi-film serves to protect the Si surface in photoelectrochemical reaction[33]. The nanowires were then grown on top of the quasi-film under nitrogen-rich conditions, leading to N-terminated surfaces[38]. The growth conditions for nanowires included a substrate temperature of ~735 °C, Ga BEP of $6 \times 10^{-8}$ torr, Si cell at 1250 °C, nitrogen flow rate of 1 sccm, forward plasma power of 350 W, and a growth duration of 4–5 h.

### $Al_2O_3$ passivation layer and Pt catalysts loading on GaN NWs

For passivation of GaN nanowires, 2 nm of $Al_2O_3$ was deposited via atomic layer deposition at 200 °C with trimethyl aluminum as precursor. Subsequent Pt cocatalysts were deposited via a routine photodeposition process[33].

## Back contact and photocathode preparation

The GaN nanowires/Si photocathode was cleaned in a 37% HCl solution for 1 – 2 min. The sample was then loaded in an electron-beam physical vapor deposition chamber to deposit a 200 nm/50 nm thick Al/Au back contact. Immediately after the depostion, the Al/Au-deposited sample was annealed for 10 min at 425 °C under ambient pressure with nitrogen gas flow in a rapid thermal annealing chamber. For the photocathode preparation, the sample was diced into pieces with areas 0.1 – 0.2 cm$^2$ using a diamond scribe. A segment of tinned copper wire was fixed to the backside of the sample with silver paste for electrical connection. The photocathode's backside and edges were sealed from the electrolyte with epoxy (Loctite EA-615).

## Photoelectrochemical measurements

PEC experiments were conducted in 0.5 M $H_2SO_4$ solution in both three-electrode and two-electrode configurations using GaN nanowires/Si photocathode, $IrO_x$, and Ag/AgCl as the working, the counter electrode, and the reference electrode, respectively. The LSV curves of the $IrO_x$ counter electrode are given in Fig. S17, from which the correspondence between the two-electrode and three-electrode LSV measurements can be made. In two-electrode measurements, 0.2 mM of Triton X-100 was also present in the electrolyte as a surfactant for enhanced hydrogen desorption, resulting in a more stable photocurrent. A solar simulator (Newport Oriel) with an AM 1.5 G filter was used as the light source. For the concentrated solar light experiment, a flow cell with Pt wire counter electrode and Ag/AgCl reference electrode was used. The adjustable solar light (1–9 suns) was illuminated on the back side of GaN NW/Si photocathode and electrical contact was made by sandwiching the Ga-In eutectic between the Cu foil and the Si wafer. Aqueous electrolyte of 0.5 M $H_2SO_4$ was continuously circulated with a rate of 28 ml/min by Masterflex Ismatec microflow pump (Cole-Parmer). $H_2$ product was stored in a tightly sealed chamber for gas chromatography (Shimadzu GC-8A).

The Faradaic efficiencies of $H_2$ evolution experiments were calculated using Eq. (1), which describes the ratio of average $H_2$ measured by gas chromatography (red dots in Figs. 1d and 3c) and $H_2$ production calculated from photocurrent (black dots in Fig. 1d and Fig. 3c):

$$\text{Faradaic Efficiency} = \frac{2 \times n_{H_2}\left(t = T_0\right) \times F}{\int_0^{T_0} I\, dt} \tag{1}$$

where $I$ is the measured photocurrent, $F$ is the Faradaic constant (96485 C/mol), and $n_{H_2}$ is the amount of $H_2$ experimentally produced for a time duration $T_0$. The $H_2$ concentration was measured with a gas chromatograph (GC, Shimadzu GC-8A) equipped with a thermal conducting detector. We performed two-electrode electrochemical impedance spectroscopy measurements at −2.3 V $vs.$ $IrO_x$ in 0.5 M $H_2SO_4$ under AM 1.5 G one sun illumination with an equivalent circuit established in previous reports[33].

## SEM and TEM characterization

SEM images were taken using Hitachi SU8000 at an accelerating voltage of 5 kV. High angle annular dark-field (HAADF)-STEM images were taken using a JEOL 2100 F microscope with a STEM aberration corrector operated at 200 kV. The GaN nanowires were mechanically transferred onto the TEM copper grid[12].

## Density functional theory (DFT) calculations

DFT calculations were performed using the generalized gradient approximation for the exchange-correlation potential. The projector augmented wave method[50,51] and a plane-wave basis set were used as implemented in the Vienna ab-initio simulation package[51]. The energy cutoff for the plane-wave basis was set to 500 eV for all relaxations and 600 eV for corresponding electronic structures computation.

A k-mesh of 13×9×1 was adopted for the primitive cell of GaN (wurtzite) $m$-plane, and the mesh density of k points was kept fixed when performing calculations related to its supercells. Symmetric slab models were adopted for all atomic surface configurations with a vacuum layer thickness of ~20 Å. In optimizing the geometric structures, van der Waals (vdW) interactions were considered by the vdW-DF level with the optB86 exchange functional (optB86-vdW)[52,53]. All structures were fully relaxed until the net force per atom was less than 0.01 eV·Å$^{-1}$. The electronic properties of N-rich GaN $m$-plane and bulk GaN were predicted with the hybrid functional (HSE06)[54,55], while the conventional GaN $m$-plane was calculated with optB86-vdW.

## XPS measurements

X-ray photoelectron spectroscopy was conducted with the Kratos Axis Ultra DLD system at Lawrence Berkeley National Laboratory. A monochromatic Al $K\alpha$ source with an incident beam energy of hυ = 1486.6 eV was used to excite the core-level electrons of the material. $Ga\,3d$, $N\,1s$, and $O\,1s$ core levels were collected, with a pass energy of 20 eV, step size of 0.05 eV, and 6 sweeps each to obtain a good signal-to-noise ratio. The survey spectrum was also acquired with a pass energy of 160 eV, a step size of 1 eV, and 3 sweeps. The measurements were performed under ultrahigh vacuum conditions (7.5 × 10$^{-9}$ torr). Spectral fitting was performed with CasaXPS. Spectral positions were calibrated with the $N\,1s$ core level binding energy peak at 397.8 eV.

## ICP-MS measurements

ICP-MS measurements were conducted using a PerkinElmer nexion 2000 ICP-MS machine with bismuth as an internal standard. The calibrations were done using the standard Ga solvent from Ultra Scientific.

## Data availability

All other data supporting the findings in this study are available from the corresponding authors upon reasonable request.

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

## Acknowledgements

The authors gratefully acknowledge research support from the Hydro-GEN Advanced Water Splitting Materials Consortium, established as part of the Energy Materials Network under the U. S. Department of Energy, under Contract Number DE-EE0008086 for the University of Michigan, and Contract Number DE-AC02-05CH11231 for the Lawrence Berkeley National Laboratory, the McGill Sustainability Systems Initiative (MSSI), and by United States Army Research Office Award W911NF2110337. The authors also acknowledge the technical support from the Lurie Nano-fabrication Facility and the Michigan Center for Materials Characterization. X. K. and H. G. thank the High-Performance Computing Center of McGill University, Calcul-Quebec, and Compute Canada for their computation facilities. The authors thank the group led by Prof. T. Hamann at the Michigan State University for providing the $IrO_x$ counter electrodes and T. Deutsch at the National Renewable Energy Laboratory for insightful discussions. This material is based upon work supported by the National Science Foundation Graduate Research Fellowship under Grant 1841052.

## Author contributions

Y.X., S.V., W.J.D., and Z.M. designed this study. X.K. and H.G. performed density functional theory calculations. S.V., Y.X., and I.A.N. prepared n +-p Si solar cell fabrication and performed MBE growth of n+-GaN nanowires on Si. S.V., W.J.D., and B.Z. contributed to sample preparation. S.V. and W.J.D. conducted photoelectrochemical and ICP-MS experiments. Y.X. prepared for the atomic layer deposition of the thin aluminum oxide passivation layer. G.Z., S.V., F.M.T., Z.Y., and Y.X. performed XPS characterization of pristine and tested n+-GaN nanowires/Si photocathode samples. Z.Y., Y.X., K.S., and S.V. performed STEM measurements on pristine and tested n+-GaN nanowires/Si photocathode samples. The manuscript was written by Y.X., X.K., S.V., W.J.D., H.G., and Z.M. with contributions from all co-authors.

## Competing interests

Some IP related to the synthesis of GaN nanowires was licensed to NS Nanotech, Inc. and NX Fuels, Inc., which were co-founded by Z. Mi. The University of Michigan and Mi have a financial interest in these companies. The remaining authors declare no competing interests.
