## [Peer Review File · Nature Communications]

Reviewers' comments:

Reviewer #1 (Remarks to the Author):

This report details a study of the use of GaN as a protecting layer for a Si water splitting system. The GaN layer indeed provides protection of the underlying Si layer. The changes in the surface of the GaN layer are assessed using photoelectron spectroscopy and an explanation of the changes in electronic structure are provided using DFT. The primary value of the study is the reported mechanistic information about the possible chemical transformations on the surface. Overall, the study is well executed but several issues should be addressed. In this reviewer's opinion, the advancement beyond the cited paper on the electrochemical stability of GaN is small. This reviewer recommends publication in a different journal such as Journal of Physical Chemistry.

1. The applied bias for operation (~ 2 V vs counter electrode for ~ 20 mA/cm²) is larger than that practically needed for a purely dark electrolysis system (1.23 V + overpotentials at the cathode/anode, perhaps 1.9 V), so although the GaN and its oxidation products result in protection of the Si, there is really nothing gained here practically. This study claims to have overcome the stability bottleneck, however, the actual performance is arguably worse than dark electrolysis.

2. The issue of surface metallization for achieving 100% faradaic efficiency is questionable; what other species in the solution can be reduced? What experimental evidence is there for surface metallization? The valence band shift could be caused by other chemical and charge effects at or near the surface. Further, how does surface metallization explain the 100% faradaic efficiency? There are many explanations for a near unity faradaic efficiency, and this high efficiency is expected as there are not other species available to be reduced in the solution.

3. The "self improving" behavior of the nitride based semiconductors occurring in 2 electrode (as well as previously reported 3 electrode) measurements is completely expected, since this is a surface effect on the semiconductor, and that aspect of the system is unchanged.

4. The change in photocurrent is not clearly explained. The photocurrent is expected to be light limited and nearly constant for the Si pn junction. Only changes in reflectance or absorbance in the GaN layer should cause changes in the photocurrent at voltages sufficiently large to be in the flat portion of the JV curve. Otherwise, changes in photocurrent would be to shifting of the JV curve, which may be possible due to changes in the surface and the catalytic behavior.

5. Statement at lines 160-162 is unclear. Why are two Von values mentioned?

6. Were the two electrodes separated via any membrane? Did the authors investigate for deposition of Ir species on the cathode surface?

7. The authors do not clearly show that the additional contributing peak for the XP spectra are indeed needed or correct. For example, could the data in fig 2b be still fit with 2 contributions? How have the authors made the assignment of the O-Ga-N contribution?

Reviewer #2 (Remarks to the Author):

This is a well-presented manuscript demonstrating long stability of Si/GaN nanowires photocathodes. The authors demonstrated this is due to the formation of an oxynitride layer at the (M-plane) surface which further improves the photocatalytic activity. DFT calculations were to support the discussion.

I recommend the manuscript to be considered for publication after the authors have adequately address the issues listed below:

- the authors argue data for 2-electrode system in the literature is scarce and hence their report here. It would be good if the authors can present the stability and performance data if their photocathodes are set-up in a 3-electrode configuration. Does the configuration affect their stability and performance? If so why?

- the XPS data was obtained at an incident angle of 60° , where m-plane surfaces of nanowires were predominantly measured. The density of the nanowires is quite high so I would expect a non-negligible contribution from the c-planes. Also all the sidewall facets the m-plane?

- unless the authors show some methodology to extract the projected 100,000 hrs lifetime, this claim should not be included. It is misleading.

- lines 187-189: "...it can be concluded that the LSV curves remained almost

identical 10 h – 3000 h due to the formation of oxynitride species on the GaN nanowires m-plane during the first 40 h of the experiment."

Where does the value 40 h come from? It is just a guess?

- the authors considered four possible atomic configurations of GaON featured GaN m-plane in the DFT calculations. How do the results related to each of this configuration? The result of only one configuration (the green one) is shown/discussed.

- the valence band of the tested sample is increased by 0.5 V compared to the pristine one. How does this value related to that calculated by DFT? In the manuscript, it was only mentioned the bandgap reduction from 3.19 eV to 1.83 eV.

Reviewer #3 (Remarks to the Author):

Vanka, Mi, et al., report on the prolonged operation of a GaN nanowire coated n+p-Si photocathode operating for > 3000 hrs in a practical two electrode configuration. They use a molecular beam epitaxy process to deposit a GaN nanowire layer on a Si buried junction photocathode, then use it to drive potential-assisted hydrogen evolution versus an IrOx water oxidation counter electrode in a simple, single cell two electrode configuration. They show that activity for hydrogen evolution actually improves over time (within 10 hours) and that the activity is stable for more than 3000 hours. The researchers then propose that the enhancement in activity and the stability is due to the formation of a Ga-O-N layer that acts as an electrocatalyst or improves the interfacial charge transfer resistance to improve the performance of the photocathode. They use XPS and DFT to support the claim that Ga-O-N clusters are formed, and that they have metallic characteristics, respectively. The researchers also spend a significant amount of the manuscript emphasizing how important two electrode experiments are for assessing the capabilities of practical photoelectrochemical water splitting systems versus three-electrode ones.

There are major deficiencies with this work that should be considered before accepting it for publication in any journal. For one, this work is extremely incremental beyond the group's recent Nature Materials paper (cited in this manuscript) that puts forward a majority of these claims already. The novelty of this specific manuscript is that the two electrode system operated for over 3000 hours. The main claim in this paper (and the previous one) is still not satisfactorily defended by the experimental results. The authors make the claim:

"We show that the in-situ formation of atomic scale GaON nanoclusters on N-terminated GaN nanowires takes place when O atoms partially replace the N atoms on the non-polar GaN m-plane. The incorporation of O atoms on GaN not only reduced surface band bending, but also created atomic-scale localized nanoclusters of semiconductor surface metallization, i.e., GaON species, which naturally act as reduction reaction sites."

It can be true that a Ga-O-N interface is formed but not even participating in the electrochemistry. The nanowire morphology is unnecessary, and obscures what transformations are actually occurring in this structure. If this was a purely one dimensional system (a simple thin film of GaN, for example), then assessing this claim would be easier. However, with the extended heterogeneous morphology, it is possible that the nanowires are forming Ga-O-N while the base layers are participating in another way, perhaps reducing to Ga metal, which would also explain the metallization of the surface. This is a possibility considering the extremely negative potentials this photocathode must be biased to in order to produce hydrogen and the Pourbaix diagram of Ga. Additionally, the authors make claims of near 100% faradaic efficiency to H₂, but omit any measurement of it before 2h into the experiment (Fig 1d). Even at 2 hours, the faradaic efficiency is only 80%. One can assume that the FE is even lower before that time and other reduction processes are occurring.

A simpler explanation of the self-healing nature of these photocathodes is that dissolved Ir from the counter electrode is plating on the photocathode after 10 hours and acting as an electrocatalyst. The researchers did not address this here, but did so in a cursory fashion in the 2021 paper after only 10 hr of operation. Again, the nanowire morphology can easily obscure the presence of Ir on the functional interface, particularly if it is at the base of these extended nanowires. The researchers knew that this would be an issue, yet completely avoided it in this manuscript. They did not use a membrane to protect the photocathode from dissolved Ir, and did not show the XPS for Ir at any point in the 3000 hour extended run. Changing the solution every 24 hours is not sufficient for avoiding this, and even measuring the surface for Ir XPS signal and seeing no appreciable signal above the level of noise (like in the 2021 Nature Materials SI) is not sufficient to say that there is no Ir on the surface.

More fundamentally, the experiments from the XPS and the conclusions are not particularly in agreement. For one, if Ga-O-N clusters are forming, the particular XPS feature corresponding to this would be extremely dim. The authors only show the O1s analysis in the main text, where the claimed Ga-O-N is on par with the Ga-O-H feature. In the SI, however, they show in the Ga 3d that the Ga-O-N is significantly more intense than the Ga-O-H feature, and of appreciable intensity compared to the bulk Ga-N peak. This is inconstant with the proposed formation of electrocatalytically-active 'clusters' on the surface (see DFT modeling), and inconsistent with these researchers own nitrogen XPS data in the 2021 paper. This is where the simple GaN starting surface would help to strengthen the conclusions or enlighten the actual important transformations occurring. More importantly, it is completely unacceptable to only present line fits (particularly where the Ga-O/Ga-O-N/Ga-O-H features are so close in energy) and do not show the actual XPS data to allow for the assessment of the fits. XPS is generally a noisy spectral measurement, and the analysis can be subjective to the fitting procedure.

There is even a significant point to the practicality of this photoelectrode. A system that requires -2.3 V of cell potential in addition to the photovoltage generated from a solid state pn+-Si junction is not a practical one for water splitting. The GaN-n+pSi photocathode shown here does not perform better than a Pt electrocathode, limiting the achievement described in this manuscript to "practical electrochemical stability in systems that are driven by electrodes that are impractical". p-Si-Pt photocathodes have been studied and shown similar stability (> 1400 h) while operating at more positive potentials than a Pt cathode. For example, see Maier, et al., Int.J. Hydrogen.Energy., 21, 859-864 (1996). While these researchers used a three-electrode system to measure the stability, it doesn't actually change the electrochemical system - they could have just as easily referenced the potential to the counterelectrode. Adding the buried junction would only improve the photovoltage, as it is decoupling the photovoltage generation from the semiconductor-liquid junction. The authors could have learned a great deal more about their system had they done three electrode characterization here. The researchers should try similar experiments with the bare pn+-Si photocathode. It would help to understand whether or not the GaN is actually a useful component in this complex photoelectrode architecture. More broadly, the authors should be careful to denigrate other approaches to photoelectrochemical characterization, namely three-electrode experiments, or two electrode experiments where the photocurrent and photovoltage are practical for realistically useful systems.

In summary, the conclusions of this paper need more well-defined experiments and subsequent characterization before it can be published. For certain, they should use a simpler GaN layer and a membrane to remove all doubt that whatever transformations are occurring at the GaN-acid interface are, in fact, electrochemically active and the only route to electrochemical activity.

Minor issues:

- When you plot PEC data (chronoamperometry, for example), you need to state what the conditions are. Fig s4 does not report the applied cell potential.
- The acronym 'MBE' is used but not spelled out.
- line 162 -- not sure what this means, or why there are two potentials cited: "photocurrent density of the sample was ~ 8 mA/cm² with V_{on} ~ -1.8 V vs. IrOx (at -2.3 V vs. IrOx)." Perhaps it is a mistake. Additionally, the characterization of V_{on} is subjective as well. While there is some small photocurrent on the '10 h' electrode at -1.35 V vs the CE, the true 'turn on' looks to be significantly more negative (-1.7 V vs CE).
- The researchers should also characterize the j-V characteristics of the pn+-Si photocathode by itself (particularly in a three electrode experiment) so readers can put the applied bias and work done by illumination into context.

- Did the researchers measure that the pH of the solution is zero, or is it assumed to be zero because of the diprotic sulphuric acid used? It is probably not pH 0.0, more like 0.4, which is a minor but important distinction.

Reviewer 1

1. The applied bias for operation (~ 2 V vs counter electrode for ~ 20 mA/cm²) is larger than that practically needed for a purely dark electrolysis system (1.23 V + overpotentials at the cathode/anode, perhaps 1.9 V), so although the GaN and its oxidation products result in protection of the Si, there is really nothing gained here practically. This study claims to have overcome the stability bottleneck, however, the actual performance is arguably worse than dark electrolysis.

Reply: Thank you for the comment. As given in Supporting Information Table S3, the saturated V_{on} in our two-electrode system, which employs no extrinsic cocatalysts whatsoever, was $-1.35\sim -1.5$ V (vs. IrO_x) under one-sun illumination throughout 3,000 hours of chronoamperometry (CA), which is better than a typical dark electrolysis system. As such, we could have easily chosen a bias less than -2.3 V for CA stability testing, but to illustrate the stability of the system more convincingly, we chose a bias that would surely operate in the saturation photocurrent regime (which exists for *pn* junction devices). After 3,000 hours of operation, the system's photoelectrochemical performance remains essentially identical to that after 100 hours, so that it is still ready to operate at desired lower biases where it would be at least comparable to dark electrolysis systems.

Significantly, the measurements were done without any extrinsic co-catalysts, e.g., noble metals such as Pt and Ir, thereby drastically reducing the overall system cost. We have also performed additional CA experiments at lower biases and confirmed that the Faradaic efficiency remained to be unity, thus confirming the viability to operate the cell at biases comparable to or lower than those used in dark electrolysis without extrinsic catalysts (Figure A1). Such unprecedented combination of desirable properties presented in this work thus represents a major breakthrough both in terms of device understanding and its practicality. In the revised manuscript, we added the newly measured Faradaic efficiency and production rate at different potentials in Figure S10 and explained the reason for choosing the bias (*i.e.*, -2.3 V vs. IrO_x) in lines 173–176.

Figure A1. Faradaic efficiency of the GaN NW/Si photoelectrode at different biases after 24 hr of chronoamperometry at -2 V vs. IrOx under one-sun illumination.

Moreover, we additionally evaluated GaN NW/Si photocathode in three-electrode configurations in 0.5 M H₂SO₄ under one sun light illumination and compared the performance with the previously reported Si photocathodes. Linear sweep voltammetry (LSV) curves of the photocathode showed *in-situ* gradual improvement of V_{on} and photocurrent density (Figure A2a). After 100 h, we measured $V_{on} = 0.36$ V vs. reversible hydrogen electrode (V_{RHE}) and photocurrent density ~ 30 mA/cm² at 0 V_{RHE} , which are among the best performances of Si photocathode without any deposited catalysts and are within the performance range of Si photocathodes with catalysts (Figure A2b). This is a significant advancement from the previous work on quasi-film GaN/Si photocathode [*Nat. Mater.* **20**, 1130–1135 (2021)] due to the unique morphology of vertically aligned GaN nanowires for which the active *m*-plane sidewalls dominate and which enhance mass transport due to increased porosity of the NW array compared to the quasi-film. In the revised manuscript, we added results of the three-electrode measurements and discussed their physical meaning (Figures 1, S2–S7, and Table S2).

Figure A2. (a) Linear sweep voltammetry (LSV) curves of the GaN NW/Si photocathode after reaction for 0 – 100 h and Pt-loaded GaN NW/Si photocathode in three-electrode configuration. (b) Performance comparison of Si photocathodes for H₂ evolution reaction. Y-axis indicates the photocurrent density at 0 V vs. reversible hydrogen electrode (V_{RHE}) and X-axis indicates the onset potential (V_{on}). *n*⁺-*p* Si photocathodes with catalysts (red circles) reveal better performance than *p*-type Si photocathodes with catalysts (black squares). Notably, GaN NW/Si (this work, highlighted blue triangle) shows a significant advancement from the previous GaN quasi-film/Si photocathode (green triangle) [*Nat. Mater.* **20**, 1130–1135 (2021)].

2. The issue of surface metallization for achieving 100% faradaic efficiency is questionable; what other species in the solution can be reduced? What experimental evidence is there for surface metallization? The valence band shift could be caused by other chemical and charge effects at or near the surface. Further, how does surface metallization explain the 100% faradaic efficiency? There are many explanations for a near unity faradaic efficiency, and this high efficiency is expected as there are not other species available to be reduced in the solution.

Reply: Thank you for the comment. We agree with the reviewer that “there are no other species available to be reduced in the solution”. As such, the *in-situ* surface oxynitride formation entirely accounts for the faradaic efficiency that is initially less than 100% but increases monotonically and finally reaches 100% (within measurement uncertainty) after 10 hours of chronoamperometry. We base the claim of surface metallization on our DFT calculations, which suggest that the Fermi level moves above the conduction band minimum at the sites of oxynitride clusters, which is consistent with the observed increase in the energy difference between the surface Fermi level and the surface valence band maximum from the XPS data. As the chronoamperometry goes beyond 10 hours, surface metallization (*i.e.* the oxynitride formation) approaches an equilibrium such that all surface electrons participate in proton reduction rather than oxynitride formation. In this regime, the Faradaic efficiency has been measured to be 100% within the measurement uncertainty.

3. The “self improving” behavior of the nitride based semiconductors occurring in 2 electrode (as well as previously reported 3 electrode) measurements is completely expected, since this is a surface affect on the semiconductor, and that aspect of the system is unchanged.

Reply: Thank you for the comment. We agree with the reviewer that the self-improving behavior of the photocathode is due to surface effects, namely the surface gallium oxynitride species formation. As explicitly stated and extensively cited in text, a three-electrode stability for a photoelectrode does not necessarily correspond to an equivalent two-electrode stability, since the three-electrode stability only indicates the stability of the working electrode at electrochemical potentials that might become irrelevant in two-electrode configurations, where the electrode-electrolyte interfaces of the working electrode and the counter electrode, as well as electrolyte conditions such as charge buildup and ionic concentration gradient, collectively and dynamically define the electrode potentials. In addition, in the previous work [*Nat. Mater.* **20**, 1130–1135 (2021)], we have grown GaN quasi-films with small portions of the nonpolar crystal planes exposed, while the non-polar surfaces are responsible for *in-situ* formation of gallium oxynitride. In this work, however, GaN NW arrays with much more predominant nonpolar sidewall exposure were tested for the self-improving effect and stability. Due to the unique morphology of vertically aligned GaN NWs for which the active *m*-plane sidewalls dominate and which enhance mass transport due to increased porosity of the NW array compared to the quasi-film, a significant advancement of performance (*i.e.*, V_{on} from -0.08 to 0.4 V_{RHE}) could be demonstrated. As such, it is highly nontrivial to extrapolate from the work presented in *Nature Materials*, which was done using GaN quasi-film/Si photocathodes under three-electrode configurations tested for a duration of 150 hr, to the 3000 hr two-electrode stability results using the GaN NW/Si presented in this work. In this context, the current report extends the state-of-the-art two-electrode stability report by one order of magnitude, which has never been demonstrated before.

4. The change in photocurrent is not clearly explained. The photocurrent is expected to be light limited and nearly constant for the Si pn junction. Only changes in reflectance or absorbance in the GaN layer should causes changes in the photocurrent at voltages sufficiently large to be in the flat portion of the JV curve. Otherwise, changes in photocurrent would be to shifting of the JV curve, which may be possible due to changes in the surface and the catalytic behavior.

Reply: Thank you for the comment. We agree with the reviewer’s assessment that “changes in photocurrent would be to shifting of the JV curve, which may be possible due to changes in the surface and the catalytic behavior”, as such “changes in the surface and the catalytic behavior” are the formation of gallium oxynitride species on the nanowire sidewalls for our photocathode. In fact, in comment 3 the reviewer stated that a surface effect which takes place on the nitride-based semiconductor (including our photocathode) leads to a self-improvement effect, *i.e.*, an increase in photocurrent at the same bias due to the improvement in onset potential.

5. Statement at lines 160-162 is unclear. Why are two V_{on} values mentioned?

Reply: Thank you for the comment. On lines 160–162, the sentence reads: “At the beginning of the stability experiment (0th hour), the photocurrent density of the sample was $\sim 8 \text{ mA/cm}^2$ with $V_{on} \sim -1.8 \text{ V vs. IrO}_x$ (at -2.3 V vs. IrO_x).” Here, the voltage cited inside the parentheses is not a V_{on} value, but rather the applied bias for the chronoamperometry. The only other voltage stated ($V_{on} \sim -1.8 \text{ V}$) is the V_{on} of the system at the beginning of the stability experiment. The sentence is reworded for greater clarity. In the revised manuscript, the sentence now reads: “At the beginning of the stability experiment (0th hour), the V_{on} was $\sim -1.8 \text{ V vs. IrO}_x$ and the photocurrent density of the sample was $\sim 8 \text{ mA/cm}^2$ at -2.3 V vs. IrO_x .” (lines 224–225)

6. Were the two electrodes separated via any membrane? Did the authors investigate for deposition of Ir species on the cathode surface?

Reply: Thank you for the comment. The photoelectrochemical measurements were taken in a single cell chamber. The XPS and TEM measurements (Figure A2) indicate that there are no Ir species on the photocathode surface. In addition, we have performed extensive studies of the photocathode in three-electrode configurations, where we find that, with a thin (2 nm) conformal layer of Al₂O₃ coated via atomic layer deposition, the photocathode no longer possesses the self-improving behavior of the bare GaN/Si photocathode (Figure A3). Moreover, such Al₂O₃-coated photocathode can still be activated with a routine platinum cocatalyst photodeposition, meaning that though the Al₂O₃ layer has passivated the GaN surface against oxynitride formation, it is thin enough for photoelectrons to tunnel through to reach the Pt cocatalysts. This control experiment demonstrates unambiguously that the Ga-O-N/electrolyte interface is essential for the self-improvement behavior of the photocathode for solar proton reduction reaction. Thus, the *in-situ* self-improvement of the GaN NW/Si photocathode does not come from possible contaminants, but from surface modification of GaN.

Figure A2. (a) Survey spectrum of X-ray photoelectron spectroscopy (XPS) of the GaN NW/Si photocathode after 48 hours of chronoamperometry at $-0.4 V_{RHE}$ under one-sun illumination. (b) XPS of the GaN NW/Si photocathode shows no Ir after 48 hours of chronoamperometry at $-0.4 V_{RHE}$ under one-sun illumination. (c) Region of interest for the Transmission electron microscopy (TEM) energy dispersive X-ray spectroscopy (EDS) spectrum is given by the box with yellow outlines and label "Area #1". (d) The TEM EDS spectrum in the region of interest shows no Ir after reaction with conditions given above.

Figure A3. (a) CA curve, (b) LSV curves, and (b) Nyquist impedance plots of Al₂O₃ (2 nm)-passivated GaN NW/Si photocathodes. There was very little increase in photocurrent density and decrease in charge transfer resistance after 24 hours of reaction at -0.4 V_{RHE} under one-sun illumination. Even after 24 hours of chronoamperometry, the negative V_{on} (< 0 V_{RHE}) and the small photocurrent density (< 10 mA/cm² at -0.4 V_{RHE}) clearly demonstrate that the performance is not as good as GaN NW/Si ($V_{on} > 0$ V_{RHE} and the photocurrent density > 30 mA/cm² at -0.4 V_{RHE}). Panel (b) also shows that, after a routine photodeposition of Pt cocatalysts, the activity of the Al₂O₃ passivated photocathode is comparable to one without Al₂O₃ passivation (after Pt deposition). These studies provide unambiguous evidence that the Ga-O-N/electrolyte interface is essential for the self-improvement behavior of the photocathode.

7. The authors do not clearly show that the additional contributing peak for the XP spectra are indeed needed or correct. For example, could the data in fig 2b be still fit with 2 contributions? How have the authors made the assignment of the O-Ga-N contribution?

Reply: Thank you for the comment. The *O 1s* peak of GaN was asymmetric due to the major O-Ga and the minor O-H bonds. However, after chronoamperometry for 24 h, the peak shape changed to be narrower and symmetrical. This narrow and symmetric *O 1s* peak cannot be fitted with the two bonding states of O-Ga and O-H, indicating that an additional bonding state of O-Ga-N was formed during chronoemperometry as we have fit the XPS peaks according to our previous work on gallium oxynitride formation [*Nat. Mater.* **20**, 1130–1135 (2021)].

[1] Zeng, G., Pham, T.A., Vanka, S. *et al.* Development of a photoelectrochemically self-improving Si/GaN photocathode for efficient and durable H₂ production. *Nat. Mater.* **20**, 1130–1135 (2021). <https://doi.org/10.1038/s41563-021-00965-w>

Reviewer 2

1. the authors argue data for 2-electrode system in the literature is scarce and hence their report here. It would be good if the authors can present the stability and performance data if their photocathodes are set-up in a 3-electrode configuration. Does the configuration affect their stability and performance? If so why?

Reply: Thank you for the comment. We are including a detailed three-electrode study in the revised manuscript (Figures 1, S2–S7), where the self-improvement is further confirmed to have originated from the surface oxynitride species formation. Under one-sun illumination, chronoamperometry (CA) curve of photocathode at -0.4 V vs. reversible hydrogen electrode (V_{RHE}) in three-electrode configuration shows a rapid increase in photocurrent density from 0.6 to ~ 35 mA/cm² over 40~50 h and stabilize thereafter (Figure A1). Under one-sun light, the exact duration at which catastrophic device failure may occur (if at all) for this photocathode is yet to be determined. Therefore, the configuration effect (two vs. three-electrode) on stability will be evaluated in harsher conditions like high temperature and concentrated sunlight to better quantify the durability of this device in future.

Figure A1. Chronoamperometry (CA) curve of the GaN NW/Si photocathode at -0.4 V vs. reversible hydrogen electrode under AM 1.5G one-sun illumination in 0.5 M H₂SO₄.

2. unless the authors show some methodology to extract the projected 100,000 hrs lifetime, this claim should not be included. It is misleading.

Reply: Thank you for the comment. We have removed the statement from our manuscript.

3. lines 187-189: "...it can be concluded that the LSV curves remained almost identical 10 h – 3000 h due to the formation of oxynitride species on the GaN nanowires m-plane during the first 40 h of the experiment." Where does the value 40 h come from? It is just a guess?

Reply: Thank you for the comment. As shown in Figures S1 and S12, the current density reached saturation somewhere between the 40- and 50-hour marks. For clarity, we have revised the value of 40 hr to a range of 40–50 hr.

4. the authors considered four possible atomic configurations of GaON featured GaN m-plane in the DFT calculations. How do the results related to each of this configuration? The result of only one configuration (the green one) is shown/discussed.

Reply: Thank you for the comment. Corresponding results for the first, third and fourth configurations have been added in the revised manuscript (Figure S16), which further support our major finding that GaON nanoclusters largely enhance the stability and the efficiency of the solar-powered artificial photosynthesis. Firstly, the formation energies for all the proposed GaON configurations are negative, which are namely -4.07 eV, -4.29 eV, -3.95 eV, and -2.66 eV. Such values indicate that the introduction of the O atom to the N-rich GaN surface is a thermodynamically favored process. Secondly, from the calculated density of states (DOS), it is found that the Fermi levels of all four GaON configurations are shifted upward into the conduction band, rendering the oxygen-incorporated surface metallized. The emerging metallic states naturally act as an electron sink and serve as atomic scale reduction reaction sites, thereby enhancing catalytic performances.

5. the valence band of the tested sample is increased by 0.5 V compared to the pristine one. How does this value related to that calculated by DFT? In the manuscript, it was only mentioned the bandgap reduction from 3.19 eV to 1.83 eV.

Reply: Thank you for the comment. Experimentally, the valence spectrum measurement for the tested sample showed an increase of ~ 0.5 V in $E_{FS} - E_{VS}$ compared to the pristine sample (Figure 3c), where E_{FS} is the surface Fermi level and E_{VS} is the surface valence band maximum. Theoretically, it is found that (Figure 5b) the Fermi levels of GaON shifted upward into the conduction band with a value of 1.39 eV after oxidization, which is qualitatively consistent with the above experimental measurement. Corresponding statement has been added in the revised main text (line 280).

[1] Zeng, G., Pham, T.A., Vanka, S. *et al.* Development of a photoelectrochemically self-improving Si/GaN photocathode for efficient and durable H₂ production. *Nat. Mater.* **20**, 1130–1135 (2021). <https://doi.org/10.1038/s41563-021-00965-w>

Reviewer 3

Since many of Reviewer 3's comments are in paragraph form, here we break down the paragraphs to respond point by point. The comments in bullet point are enumerated afterwards and responded point by point as well.

P1. Vanka, Mi, et al., report on the prolonged operation of a GaN nanowire coated n+p-Si photocathode operating for > 3000 hrs in a practical two electrode configuration. They use a molecular beam epitaxy process to deposit a GaN nanowire layer on a Si buried junction photocathode, then use it to drive potential-assisted hydrogen evolution versus an IrOx water oxidation counter electrode in a simple, single cell two electrode configuration. They show that activity for hydrogen evolution actually improves over time (within 10 hours) and that the activity is stable for more than 3000 hours. The researchers then propose that the enhancement in activity and the stability is due to the formation of a Ga-O-N layer that acts as an electrocatalyst or improves the interfacial charge transfer resistance to improve the performance of the photocathode. They use XPS and DFT to support the claim that Ga-O-N clusters are formed, and that they have metallic characteristics, respectively. The researchers also spend a significant amount of the manuscript emphasizing how important two electrode experiments are for assessing the capabilities of practical photoelectrochemical water splitting systems versus three-electrode ones.

Reply: We thank the reviewer for the summary description of the work in this paragraph (paragraph 1).

P2a. There are major deficiencies with this work that should be considered before accepting it for publication in any journal. For one, this work is extremely incremental beyond the group's recent Nature Materials paper (cited in this manuscript) that puts forward a majority of these claims already. The novelty of this specific manuscript is that the two electrode system operated for over 3000 hours. The main claim in this paper (and the previous one) is still not satisfactorily defended by the experimental results.

Reply: Thank you for the comment. *As explicitly stated and extensively cited in text, a three-electrode stability for a photoelectrode does not necessarily correspond to an equivalent two-electrode stability, since the three-electrode stability only indicates the stability of the working electrode at electrochemical potentials that might become irrelevant in two-electrode configurations, where the electrode-electrolyte interfaces of the working electrode and the counter electrode, as well as electrolyte conditions such as charge buildup and ionic concentration gradient, collectively and dynamically define the electrode potentials. As such, it is highly nontrivial to extrapolate from the work presented in Nature Materials [Nat. Mater. 20, 1130–1135 (2021)], which was done under three-electrode configurations tested for a duration of 150 hr, to the 3000 hr two-electrode stability results presented in this work from first principles. In this context, the current report extends the state-of-the-art two-electrode stability report by one order of magnitude, which has never been achieved in any material system before.*

Moreover, the photocathode presented in this work employs a nanowire (NW) morphology, which allows for much greater exposure of the non-polar sidewalls to the electrolyte than the quasi-film morphology studied previously. With additional three-electrode experiments, we showed that the GaN NW/Si photocathode can achieve much more positive onset potential than the GaN quasi-film on Si photocathode, meaning that the self-improvement effect on the former is much greater than on the latter due to much greater non-polar sidewall exposure. GaN NW/Si exhibited $V_{on} = 0.36$ V vs reversible hydrogen electrode (V_{RHE}) and photocurrent density = ~ 30 mA/cm² at 0 V_{RHE} are among the best performances of Si photocathode without deposited catalysts and are within the performance range of Si photocathodes with catalysts (Figure A1). In the revised manuscript, we added results of three-electrode configuration and discussed their physical meaning (Figures 1, S2 – S7, and Table S2).

Figure A1. Performance comparison of Si photocathodes for H₂ evolution reaction. The Y-axis indicates the photocurrent density at 0 V vs. reversible hydrogen electrode (V_{RHE}) and the X-axis indicates the onset potential (V_{on}). n^+p Si photocathodes with catalysts (red circles) reveal better performance than p -type Si photocathodes with catalysts (black squares). Notably, GaN NW/Si (this work, highlighted blue triangle) exhibits a great advance from the previous GaN quasi-film/Si photocathode (green triangle) [*Nat. Mater.* **20**, 1130–1135 (2021)] due to the beneficial nanowire morphology of GaN.

P2b. The authors make the claim: "We show that the in-situ formation of atomic scale GaON nanoclusters on N-terminated GaN nanowires takes place when O atoms partially replace the N atoms on the non-polar GaN m-plane. The incorporation of O atoms on GaN not only reduced surface band bending, but also created atomic-scale localized nanoclusters of semiconductor surface metallization, i.e., GaON species, which naturally act as reduction reaction sites." It can be true that a Ga-O-N interface is formed but not even participating in the electrochemistry.

Reply: Thank you for the comment. We have performed additional studies of the photocathode in three-electrode configurations, where we find that, with a thin (2 nm) conformal layer of Al₂O₃ coated via atomic layer deposition, the photocathode no longer possesses the self-improving behavior of the bare GaN/Si

photocathode (Figure A2). Moreover, such Al₂O₃-coated photocathode can still be activated with a routine platinum cocatalyst photodeposition, meaning that though the Al₂O₃ layer has passivated the GaN surface against oxynitride formation, it is thin enough for photoelectrons to tunnel through to reach the Pt cocatalysts. This control experiment demonstrates unambiguously that the Ga-O-N/electrolyte interface is essential for the self-improvement behavior of the photocathode for solar proton reduction reaction.

Figure A2. (a) CA curve, (b) LSV curves, and (b) Nyquist impedance plots of Al₂O₃ (2 nm)-passivated GaN NW/Si photocathodes. There was very little increase in photocurrent density and decrease in charge transfer resistance after 24 hours of reaction at -0.4 V_{RHE} under one-sun illumination. Even after 24 hours of chronoamperometry, the negative V_{on} (< 0 V_{RHE}) and the small photocurrent density (< 10 mA/cm² at -0.4 V_{RHE}) clearly demonstrate that the performance is not as good as GaN NW/Si ($V_{on} > 0$ V_{RHE} and the photocurrent density > 30 mA/cm² at -0.4 V_{RHE}). Panel (b) also shows that, after a routine photodeposition of Pt cocatalysts, the activity of the Al₂O₃ passivated photocathode is comparable to one without Al₂O₃ passivation (after Pt deposition). These studies provide unambiguous evidence that the Ga-O-N/electrolyte interface is essential for the self-improvement behavior of the photocathode.

P2c. The nanowire morphology is unnecessary, and obscures what transformations are actually occurring in this structure. If this was a purely one dimensional system (a simple thin film of GaN, for example), then assessing this claim would be easier. However, with the extended heterogeneous morphology, it is possible that the nanowires are forming Ga-O-N while the base layers are participating in another way, perhaps reducing to Ga metal, which would also explain the metallization of the surface.

Reply: Thank you for the comment. We have previously found that gallium nitride films, wherein the polar c -plane dominates, grown on silicon by themselves have essentially no activity for photoelectrochemical proton reduction, providing unambiguous evidence that the exposure of non-polar sidewalls (instead of the base layers mentioned by the reviewer) is essential [*Nano Lett.* 2022, 22, 6, 2236–2243]. Then, one way to enlarge the portion of non-polar sidewalls is the growth of GaN nanowire arrays along the c -axis. In this work, we successfully demonstrated vertically aligned GaN NW array on Si photocathode and showed, for the first time, dramatically improved activity for hydrogen evolution reaction compared to the previously reported GaN quasi-film/Si photocathode.

P2d. This is a possibility considering the extremely negative potentials this photocathode must be biased to in order to produce hydrogen and the Pourbaix diagram of Ga.

Reply: Thank you for the comment. The bias of -2.3 V was chosen to ensure that the photocathode would operate in the saturated reverse biased photocurrent regime for the majority of the 3000 hr stability run. We have confirmed with additional experiments that the hydrogen evolution Faradaic efficiency remains unity at various less negative biases (Figure A3).

Figure A3. Faradaic efficiency of the GaN NW/Si photoelectrode at different biases after 24 hr of chronoamperometry at -2 V vs. IrO_x under one-sun illumination.

P2e. Additionally, the authors make claims of near 100% faradaic efficiency to H₂, but omit any measurement of it before 2h into the experiment (Fig 1d). Even at 2 hours, the faradaic efficiency is only 80%. One can assume that the FE is even lower before that time and other reduction processes are occurring.

Reply: Thank you for the comment. The less than unity Faradaic efficiency in the initial hours of chronoamperometry is another piece of evidence for the formation of oxynitride on the photocathode (in the initial hours of the chronoamperometry run). As can be seen in Fig. 3b, the linear sweep voltammogram (LSV) of the photocathode after 10 hours of chronoamperometry shows drastically improved reduction kinetics compared to the photocathode before the chronoamperometry (at 0hr). The consumption of electrons for the formation of oxynitride on the photocathode surface is responsible for the fact that the Faradaic efficiency is less than unity initially but increases steadily to unity after 10 hours of chronoamperometry, by which time the photocathode surface oxynitride formation has largely reached an equilibrium, as the 10-hr LSV of the photocathode is very similar to the LSV curves at much later stages in the 3,000 hr chronoamperometry. As part of the detailed three-electrode study included in the revised manuscript, we have measured the Faradaic efficiency at smaller intervals, showing that the Faradaic efficiency increased from 52% to unity during the initial hours (Figure A4).

Figure A4. Faradaic efficiency of H₂ measured at -0.4 V_{RHE} for initial 4 h and after CA test (25 h). The measurements were taken at -0.4 V_{RHE} under AM 1.5G 1 sun light illumination.

P3. A simpler explanation of the self-healing nature of these photocathodes is that dissolved Ir from the counter electrode is plating on the photocathode after 10 hours and acting as an electrocatalyst. The researchers did not address this here, but did so in a cursory fashion in the 2021 paper after only 10 hr of operation. Again, the nanowire morphology can easily obscure the presence of Ir on the functional interface, particularly if it is at the base of these extended nanowires. The researchers knew that this would be an issue, yet completely avoided it in this manuscript. They did not use a membrane to protect the photocathode from dissolved Ir, and did not show the XPS for Ir at any point in the 3000 hour extended run. Changing the solution every 24 hours is not sufficient for avoiding this, and even measuring the surface for Ir XPS signal and seeing no appreciable signal above the level of noise (like in the 2021 Nature Materials SI) is not sufficient to say that there is no Ir on the surface.

Reply: Thank you for the comment. Additional XPS and TEM EDS measurements show that Ir is indeed not on the surface of the photocathode (Figure A5). We also would like to mention that if the Ir deposition were responsible for self-improvement, the Al₂O₃ layer-passivated GaN NW/Si device discussed above in reply to P2b should also show self-improvement during the reaction. However, there was not any noticeable improvement after 24 hours of reaction at -0.4 V_{RHE} under one-sun illumination (Figure A2). These extensive characterization and control experiments provide unambiguous evidence that the *in-situ* self-improvement of GaN NW/Si does not come from contaminants, but from surface modification of GaN.

Figure A5. (a) Survey spectrum of X-ray photoelectron spectroscopy (XPS) of the GaN NW/Si photocathode after 48 hours of chronoamperometry at $-0.4 V_{RHE}$ under one-sun illumination. (b) XPS of the GaN NW/Si photocathode shows no Ir after 48 hours of chronoamperometry at $-0.4 V_{RHE}$ under one-sun illumination. (c) Region of interest for the Transmission electron microscopy (TEM) energy dispersive X-ray spectroscopy (EDS) spectrum is given by the box with yellow outlines and label “Area #1”. (d) The TEM EDS spectrum in the region of interest shows no Ir after reaction with conditions given above.

P4a. More fundamentally, the experiments from the XPS and the conclusions are not particularly in agreement. For one, if Ga-O-N clusters are forming, the particular XPS feature corresponding to this would be extremely dim. The authors only show the O1s analysis in the main text, where the claimed Ga-O-N is on par with the Ga-O-H feature. In the SI, however, they show in the Ga 3d that the Ga-O-N is significantly more intense than the Ga-O-H feature, and of appreciable intensity compared to the bulk Ga-N peak. This is inconstant with the proposed formation of electrocatalytically-active 'clusters' on the surface (see DFT modeling), and inconsistent with these researchers own nitrogen XPS data in the 2021 paper.

Reply: Thank you for the comment. With due respect, we would like to point out that there is no such Ga-O-H peak in the *Ga 3d* XPS spectrum in the figures we have shown. In terms of consistency between the different orbital spectra, we would also like to point out the relative intensity of the Ga-O peak, which is the only signal present in both the *O 1s* and the *Ga 3d* spectrum that we have shown, is reduced in both spectra. Furthermore, the XPS is a technique where the surface features, such as partial replacement of the nitrogen atoms by oxygen atoms, should be prominent. Lastly, our previous work [*Nat. Mater.* **20**, 1130–1135 (2021)] suggests that the nitrogen *N 1s* spectrum is largely unaffected by the surface oxynitride formation. As such, the *N 1s* spectra have been used as energy calibration in this work as well as previously [*Nat. Mater.* **20**, 1130–1135 (2021)].

P4b. This is where the simple GaN starting surface would help to strengthen the conclusions or enlighten the actual important transformations occurring.

Reply: Thank you for the comment. As stated in reply to **P2c**, we have previously found that gallium nitride films (which are grown on silicon as well), where the polar *c*-plane dominates, by themselves have essentially no activity for photoelectrochemical proton reduction, indicating that the exposure of non-polar sidewalls is essential [*Nano Lett.* 2022, 22, 6, 2236–2243].

P4c. More importantly, it is completely unacceptable to only present line fits (particularly where the Ga-O/Ga-O-N/Ga-O-H features are so close in energy) and do not show the actual XPS data to allow for the assessment of the fits. XPS is generally a noisy spectral measurement, and the analysis can be subjective to the fitting procedure.

Reply: Thank you for the comment. We have now included the raw XPS intensity data in the revised figures.

P5a. There is even a significant point to the practicality of this photoelectrode. A system that requires -2.3 V of cell potential in addition to the photovoltage generated from a solid state pn+-Si junction is not a practical one for water splitting. The GaN-n+pSi photocathode shown here does not perform better

than a Pt electrocathode, limiting the achievement described in this manuscript to "practical electrochemical stability in systems that are driven by electrodes that are impractical". p-Si-Pt photocathodes have been studied and shown similar stability (> 1400 h) while operating at more positive potentials than a Pt cathode. For example, see Maier, et al., *Int.J. Hydrogen.Energy.*, 21, 859-864 (1996). While these researchers used a three-electrode system to measure the stability, it doesn't actually change the electrochemical system - they could have just as easily referenced the potential to the counterelectrode.

Reply: Thank you for the comment. For reasons given above in reply to **P2e**, a three-electrode stability test is not comparable with a two-electrode one. In particular, in a three-electrode system, the working electrode's potential is measured and controlled with respect to the reference electrode by driving the counter electrode at whatever potential necessary. As such, three-electrode configuration masks instabilities in the whole cell by driving the counter electrode with greater and greater bias while the working electrode-electrolyte interface maintains similar levels of activity. The difference between three-electrode and two-electrode stability has already been documented and summarized in Table S1 in the Supplementary Information. In the table, we can see that conventional high efficiency III-V photoelectrodes show reasonable stability in three-electrode configurations, but the stability is reduced by orders of magnitude when tested in two-electrode configurations.

Additionally, the saturated V_{on} in our two-electrode system, which employs no extrinsic cocatalysts whatsoever, was -1.35~-1.5 V (*vs.* IrOx) throughout 3,000 hours of chronoamperometry, which is better than a typical dark electrolysis system. As such, as given in reply to **P2d** and Figure A3, we could have easily chosen a bias less than -2.3 V for chronoamperometry stability testing, but to illustrate the stability of the system more convincingly, we chose a bias that would surely operate in the saturation current regime (which exists for *pn* junction devices). After 3,000 hours of operation with barely any maintenance in between, the system's photoelectrochemical performance remained essentially identical to that after 100 hours, so that it is still ready to operate at desired lower biases where it would be at least comparable to dark electrolysis systems, all the while remaining free of extrinsic co-catalysts, such as noble metal Pt or Ir, thereby drastically reducing the overall system costs. We have also performed additional chronoamperometry experiments at lower biases and confirmed that the Faradaic efficiency remained to be unity within measurement uncertainties, thus confirming the viability to operate the cell biases comparable or lower than those used in dark electrolysis, all the while remaining free of extrinsic catalysts. Such unprecedented combination of desirable properties presented in this work thus represents a major breakthrough both in terms of device understanding and its practicality.

Moreover, in the newly measured LSV curves in three-electrode configurations, GaN NW/Si after 100 hours showed $V_{on} = 0.36$ V *vs* reversible hydrogen electrode (V_{RHE}) and photocurrent density = ~ 30 mA/cm² at 0 V_{RHE} (Figure A6), which has never been demonstrated by electrocatalysis in dark or photoelectrocatalysis without extrinsic catalysts. Thus, the photovoltage generated from GaN NW/Si and the catalytic activity of gallium oxynitride demonstrated unprecedented performance for hydrogen evolution reaction.

Figure A6. LSV curves of the GaN NW/Si photocathode after reaction for 0 – 100 h and Pt-loaded GaN NW/Si photocathode in 0.5 M H₂SO₄ under one-sun light illumination.

P5b. Adding the buried junction would only improve the photovoltage, as it is decoupling the photovoltage generation from the semiconductor-liquid junction.

Reply: Thank you for the comment. We agree with this assessment. We would further like to point out that the gallium nitride nanowires on top of such a buried silicon junction serves as a multi-purpose protection layer, as discussed in our previous work [*Nano Lett.* 2018, 18, 6530–6537].

P5c. The authors could have learned a great deal more about their system had they done three electrode characterization here.

Reply: Thank you for the comment. As detailed in reply to **P2b**, we have included additional three-electrode characterization in the main text and the Supplementary Information that conclusively shows the role oxynitride formation on the surface plays in the self-improvement effect.

P5d. The researchers should try similar experiments with the bare pn⁺-Si photocathode. It would help to understand whether or not the GaN is actually a useful component in this complex photoelectrode architecture.

Reply: Thank you for the comment. Our previous work has demonstrated the difference in photoelectrochemical hydrogen evolution activity between a *p-n*⁺ Si photocathode and one with the GaN nanowire protection layer [*Nano Lett.* 2018, 18, 6530–6537]. In short, without GaN nanowires, the bare Si photocathode exhibits very poor performance and very poor stability, as commonly reported.

P5e. More broadly, the authors should be careful to denigrate other approaches to photoelectrochemical characterization, namely three-electrode experiments, or two electrode experiments where the photocurrent and photovoltage are practical for realistically useful systems.

Reply: Thank you for the comment. We have included results on our photocathode in three-electrode experiments and compared our results to other three-electrode experiments in the literature (Figure S3 and

Table S2). We have already compiled extensively the two-electrode stability results in the literature as well (Table S1). Currently, there simply is not any other two-electrode system that approaches the stability presented in this work. We would be grateful if the reviewer can help identify any other reports that we may have missed.

P6. In summary, the conclusions of this paper need more well-defined experiments and subsequent characterization before it can be published. For certain, they should use a simpler GaN layer and a membrane to remove all doubt that whatever transformations are occurring at the GaN-acid interface are, in fact, electrochemically active and the only route to electrochemical activity.

Reply: Thank you for the comment. We have conducted additional three-electrode experiments on our photocathode which have confirmed that the self-improvement effect is indeed due to the surface transformation at the GaN/electrolyte interface, which can be (a) blocked with passivation layers such as a thin (2 nm) layer of Al₂O₃ (Figure S7) and (b) expediated with high intensity illumination (Figures 1c, 1d, and S9). As given in reply to **P3**, we have also performed XPS and TEM EDS measurements and confirmed that there is no Ir contamination on the nanowires (Figure S8). As given in reply to **P2c**, we have found that a GaN film with little to no nonpolar sidewalls has essentially no photoelectrochemical (PEC) hydrogen evolution activity [*Nano Lett.* 2022, 22, 6, 2236–2243]. Due to the necessity of non-polar sidewalls for (hydrogen evolution on) GaN, a nanowire morphology is necessary.

1. When you plot PEC data (chronoamperometry, for example), you need to state what the conditions are. Fig s4 does not report the applied cell potential.

Reply: Thank you for the comment. We have now stated the applied cell potential in the caption.

2. The acronym 'MBE' is used but not spelled out.

Reply: Thank you for the comment. The acronym “MBE” stands for “molecular beam epitaxy” and it has now been spelled out in text.

3. line 162 -- not sure what this means, or why there are two potentials cited: "photocurrent density of the sample was ~ 8 mA/cm² with V_{on} ~ -1.8 V vs. IrO_x (at -2.3 V vs. IrO_x)." Perhaps it is a mistake. Additionally, the characterization of V_{on} is subjective as well. While there is some small photocurrent on the '10 h' electrode at -1.35 V vs the CE, the true 'turn on' looks to be significantly more negative (-1.7 V vs CE).

Reply: Thank you for the comment.

(a) The voltage cited inside the parentheses (-2.3 V vs. IrO_x) is the applied bias for the chronoamperometry. The other voltage stated ($V_{on} \sim -1.8$ V) is the V_{on} of the system at the beginning of the stability experiment. The sentence is reworded for greater clarity. In the revised manuscript, the sentence now reads: “At the beginning of the stability experiment (0th hour), the V_{on} was ~ -1.8 V vs. IrO_x and the photocurrent density of the sample was ~ 8 mA/cm² at -2.3 V vs. IrO_x.” (lines 224–225)

(b) We have stated in text that that V_{on} corresponds to the voltage whereby the photocurrent density reaches 1 mA/cm^2 . We would like to emphasize again that the photocathode has no foreign co-catalysts deposited and instead relies on the *in-situ* formation of native surface gallium oxynitride for improvement in surface catalytic properties. As such, the turn-on in the j-V curves is unsurprisingly not very sharp, and so the definition of V_{on} as the voltage whereby the photocurrent density reaches 1 mA/cm^2 is as valid as any other definition of V_{on} here.

4. The researchers should also characterize the j-V characteristics of the pn⁺-Si photocathode by itself (particularly in a three electrode experiment) so readers can put the applied bias and work done by illumination into context.

Reply: Thank you for the comment. The j-V characteristics of the pn^+ -Si photocathode with platinum cocatalyst has been characterized in our previous work [*Nano Lett.* 2018, 18, 6530–6537], where we see very poor fill factor indicating a sluggish surface kinetics, whereas the GaN NW/Si photoelectrode shows drastically improved kinetics.

5. Did the researchers measure that the pH of the solution is zero, or is it assumed to be zero because of the diprotic sulphuric acid used? It is probably not pH 0.0, more like 0.4, which is a minor but important distinction.

Reply: Thank you for the comment. We have now measured the pH of the solution, and it is 0.16. This value does not change any conclusions made in the text. We have updated the value accordingly.

[1] Zeng, G., Pham, T.A., Vanka, S. *et al.* Development of a photoelectrochemically self-improving Si/GaN photocathode for efficient and durable H_2 production. *Nat. Mater.* **20**, 1130–1135 (2021). <https://doi.org/10.1038/s41563-021-00965-w>

[2] Xiao, Y., Vanka, S., Pham, T.A. *et al.* Crystallographic Effects of GaN Nanostructures in Photoelectrochemical Reaction. *Nano Lett.* 2022, 22, 6, 2236–2243

[3] Vanka, S. *et al.* High Efficiency Si Photocathode Protected by Multifunctional GaN Nanostructures. *Nano Lett.* 2018, 18, 6530–6537

REVIEWERS' COMMENTS

Reviewer #2 (Remarks to the Author):

The authors have adequately addressed all the concerns raised by me.

Reviewer #3 (Remarks to the Author):

I have reviewed this manuscript both before and after the revisions. There are a few points that the authors have satisfied my initial issues. I find the evidence convincing there is an electrocatalyst on the surface that is not Ir plating from the counter electrode. They also used a thin Al₂O₃ layer to passivate the electrode and show that there is a transformation on the surface that leads to that. I appreciate the authors following up on those points, as it is an important control experiment for their results.

However, the main issues of novelty and utility remain. While the authors still believe that two electrode stability is the ultimate metric for EC stability, if they choose a stable counterelectrode (which they presume IrO_x is in their two electrode experiments), then the three electrode experiment is an equivalent measurement. It is, in fact, a measurement of the stability of the photocathode at an absolute potential, regardless of behavior counterelectrode. If the three-electrode experiment shows stability for an extended period of time but the two electrode experiment does not, it is an issue with the counterelectrode or the membrane failing (when one is used). Given the stability of a photocathode at an absolute operating potential, it is rather easy to extrapolate the stability of the photocathode in a two electrode system based on three electrode measurements when paired with an appropriately well-characterized, stable anode. The three-electrode experiment is actually important for knowing the operating point of each electrode, which as the authors note, is independent of whatever is occurring on the counterelectrode. In a previous review, I noted one example of a similarly durable photocathode based on Pt-pSi, which also had a voltage savings compared to a dark cathode.

The authors added some three electrode experiments to the revised draft (Figure 1). These measurements in conflict with the two-electrode experiments in Figure 4b. The three electrode measurements show improvement in the onset potential over time to onset potentials positive of HER, while the two electrode experiment seems to still require > 1.5 V vs IrO_x before turn on. That

should be roughly $\sim 1.2\text{V}$ vs IrOx after 100 hrs if the three-electrode experiment is considered. These measurements are inconsistent with each other.

These two main points (the two electrode system stability and the onset potential for the photocathode) are the main points of novelty claimed by the authors. The authors have clearly put a significant amount of work into revising this manuscript, but it still an incremental advance. Many of the issues from prior reviews are still valid.

Reviewer #4 (Remarks to the Author):

Authors demonstrate substantial achievement of long-term two-electrode operation of GaN/Si photoelectrochemical water splitting without any metallic catalyst. I deem the manuscript to be innovative and appealing to audience of Nat. Communications. The first round of revisions and replies by authors has considerably improved and clarified the content and impact of the work. Regarding the computational part I have these remarks:

1) In what respects are presented DFT calculations different to previous DFT work on this materials [Nat. Mater. 20, 1130–1135 (2021), and other DFT references are suggested to be referenced to].

2) p.13 : The authors claim to underpin the long-term photoelectrochemical stability based on DFT calculations with the statements "The negative formation energy indicates that the introduction of the O atom to the N-rich GaN surface is a thermodynamically favored process, which also agrees with the consensus that metal-O bonds are stronger than metal-N bonds. The stability of Ga-O-N species was therefore verified." For general audience, this statements/reasoning should be deepened in text or SI file. Are DFT calculations herein relevant to estimate stability of GaON in aqueous solution at all?

3) Table S4 : what is the precision of the values of formation energy?

4) Authors used k-mesh 13x9x1 for primitive GaN cell m-plane. Was convergence of DOS investigated with respect to k-mesh quality?

5) Reader would acknowledge if the 1.39eV shift in the conduction band would be marked on Fig. 5(b). Was band diagram in Fig.5e sketched or calculated, based on which assumptions?

Reviewer 2

The authors have adequately addressed all the concerns raised by me.

Reply: We thank the reviewer again for their valuable comments that have led to improvements for this manuscript.

Reviewer 3

1. I have reviewed this manuscript both before and after the revisions. There are a few points that the authors have satisfied my initial issues. I find the evidence convincing there is an electrocatalyst on the surface that is not Ir plating from the counter electrode. They also used a thin Al₂O₃ layer to passivate the electrode and show that there is a transformation on the surface that leads to that. I appreciate the authors following up on those points, as it is an important control experiment for their results.

Reply: We thank the reviewer again for their valuable comments that have led to improvements for this manuscript.

2. However, the main issues of novelty and utility remain. While the authors still believe that two electrode stability is the ultimate metric for EC stability, if they choose a stable counterelectrode (which they presume IrOx is in their two electrode experiments), then the three electrode experiment is an equivalent measurement. It is, in fact, a measurement of the stability of the photocathode at an absolute potential, regardless of behavior counterelectrode. If the three-electrode experiment shows stability for an extended period of time but the two electrode experiment does not, it is an issue with the counterelectrode or the membrane failing (when one is used). Given the stability of a photocathode at an absolute operating potential, it is rather easy to extrapolate the stability of the photocathode in a two electrode system based on three electrode measurements when paired with an appropriately well-characterized, stable anode. The three-electrode experiment is actually important for knowing the operating point of each electrode, which as the authors note, is independent of whatever is occurring on the counterelectrode. In a previous review, I noted one example of a similarly durable photocathode based on Pt-pSi, which also had a voltage savings compared to a dark cathode.

Reply: Thank you for the comment. While we agree with the reviewer that three-electrode characterizations are irreplaceable in providing fundamental understanding of the photoelectrodes, references compiled in Table S1 in the Supplementary Information provided unambiguous evidence that, for various high-efficiency device structures, the two-electrode and three-electrode stability can be very different, with the two-electrode stability often much less than the respective three-electrode one. To discuss one (out of the many cited in Table S1) sample reference here explicitly, the photoelectrode presented in [*Nat. Energy* **2**, 17028 (2017)] is much more stable under a three-electrode condition than the two-electrode condition that is similarly in the reverse bias saturation current density regime, as evident in their Supplementary Fig. 7, which is reproduced below as **Figure A1**. Like other references listed in Table S1, the (photo)electrochemical setup here has been carefully designed, such that the dramatic discrepancy between two-electrode and three-electrode stability cannot be simply assigned to counter electrode instability. A three-electrode chronoamperometry is fundamentally different from a two-electrode one because, in the former, the potential of the working electrode is held

explicitly constant with respect to its surrounding electrolyte (given that a stable reference electrode is employed), while in the latter (*i.e.*, a two-electrode chronoamperometry), no such *explicit* reference can possibly be made. Put differently, in a two-electrode chronoamperometry, both the working electrode's and the counter electrode's potentials with respect to the electrolyte are variables (rather than input) of the system, since only the difference between the potentials (at the two electrodes) is input (*i.e.*, being *explicitly* defined). Especially given the **dynamic nature of self-improving surface modification** that the photoelectrode presented in our work undergoes, the stability of the photocathode with native catalysts under two-electrode chronoamperometry cannot be inferred from previous three-electrode chronoamperometry for a photocathode with similar material design but different surface morphology (nanowire vs. quasi-epilayer) [*Nat. Mater.* **20**, 1130–1135 (2021)].

Since the stability of the Pt/*p*-Si photocathode that the reviewer referred to [*Int.J. Hydrogen.Energy.*, **21**, 859-864 (1996)] during the last round of comments was measured under three-electrode conditions, it is not comparable with the two-electrode stability results presented in this manuscript for reasons given above. In addition, this Pt/*p*-Si photocathode exhibited a low photocurrent density of ~ 4 mA/cm² under less intense light illumination (33 mW/cm²) and shorter stability testing duration (60 days = 1440 hours) compared to the GaN NW/Si photocathode evaluated here under much harsher reaction conditions (photocurrent density ~ 30 mA/cm² under 100 mW/cm² light) for 3000 hours. It should be noted that the external Pt catalysts decorated on photoelectrodes are known to degrade more rapidly under harsh reaction conditions due to detachment from the support as previously reported [*J. Mater. Chem. A* **7**, 27612 (2019), *ACS Appl. Mater. Interfaces* **7**, 18560 (2015), and *Adv. Energy Mater.* **8**, 1802585 (2018)].

Figure A1. Comparison between the two-electrode vs three-electrode stability under similar reverse-bias saturation current regime for the device presented in [*Nat. Energy* **2**, 17028 (2017)], from which this figure is reproduced.

3. The authors added some three electrode experiments to the revised draft (Figure 1). These measurements in conflict with the two-electrode experiments in Figure 4b. The three electrode measurements show improvement in the onset potential over time to onset potentials positive of HER, while the two electrode experiment seems to still require > 1.5 V vs IrO_x before turn on. That should be roughly ~ 1.2 V vs IrO_x after 100 hrs if the three-electrode experiment is considered. These measurements are inconsistent with each other.

Reply: Thank you for the comment. As the reviewer commented, the three-electrode measurements showed improvement in the onset potential (V_{on}) over time to ~ 0.4 V vs. reversible hydrogen electrode (V_{RHE}) and the two-electrode measurements exhibited V_{on} of ~ 1.5 V vs. IrO_x after improvement. V_{on} in the two-electrode measurements can be estimated from the difference between the V_{on} of oxygen evolution reaction (OER) at the anode and the V_{on} of hydrogen evolution reaction (HER) at photocathode by the following equation:

$$V_{on(2\text{-electrode})} = V_{on(3\text{-electrode OER of IrO}_x)} - V_{on(3\text{-electrode HER of GaN NW/Si})}$$

It should be noted that V_{on} has been defined in text as the potential corresponding to the current density of 1 mA/cm^2 . Therefore, the evaluation of OER performance is necessary to confirm the consistency between three-electrode and two-electrode linear sweep voltammetry (LSV) measurements. In order to check the OER performance of the IrO_x anode, we measured LSV curves in $0.5 \text{ M H}_2\text{SO}_4$ with a three-electrode configuration (Figure A2). The average value of $V_{on(3\text{-electrode OER of IrO}_x)}$ was $\sim 1.9 V_{RHE}$. Considering $V_{on(3\text{-electrode HER of GaN NW/Si})} \sim 0.4 V_{RHE}$, the calculated $V_{on(2\text{-electrode})}$ was ~ 1.5 V vs IrO_x. Therefore, the results of the three-electrode measurements are consistent with the two-electrode measurements. The LSV curves of the IrO_x counter electrode have been added to the Supplementary Information to aid readers in drawing the correspondence between the three-electrode and two electrode LSV curves.

Figure A2. Linear sweep voltammetry curves of IrO_x anode measured in $0.5 \text{ M H}_2\text{SO}_4$. Three individual measurements were performed and the averaged onset potential at 1 mA/cm^2 was $\sim 1.9 V_{RHE}$.

Reviewer 4

1. In what respects are presented DFT calculations different to previous DFT work on this materials [Nat. Mater. 20, 1130–1135 (2021), and other DFT references are suggested to be referenced to].

Reply: We thank the reviewer for bringing up this question. We explain as follows. The DFT work in [*Nat. Mater.* 20, 1130–1135 (2021)] (Fig.4 and Fig.SI-10) mainly showcases the geometric structure and formation energy calculations of O-doping to demonstrate the stability of the GaON structure.

In this manuscript, we have focused on revealing the surface metallization induced by GaON species formation on GaN nanowires, particularly their electronic structure and physical mechanisms for the enhanced catalytic performances. This atomic-scale surface metallization underlies the experimental achievement of overcoming the traditional necessity of extrinsic cocatalysts and the stability bottleneck of semiconductor photoelectrodes, offering a path for practical application of photoelectrochemical devices and systems for clean energy.

2. p.13 : The authors claim to underpin the long-term photoelectrochemical stability based on DFT calculations with the statements "The negative formation energy indicates that the introduction of the O atom to the N-rich GaN surface is a thermodynamically favored process, which also agrees with the consensus that metal-O bonds are stronger than metal–N bonds. The stability of Ga-O-N species was therefore verified." For general audience, this statements/reasoning should be deepened in text or SI file. Are DFT calculations herein relevant to estimate stability of GaON in aqueous solution at all?

Reply: Thanks for the suggestion, and in the revision, we have appended the descriptions to the caption of Table S6 in the Supporting Information as follows: In general, metal-O bonds are stronger than metal-N bonds. This is because oxygen is more electronegative than nitrogen, meaning that it attracts electrons more strongly. As a result, the metal-O bond is more polarized and the oxygen atom carries a partial negative charge, while the metal carries a partial positive charge. This polarization leads to a stronger bond. Additionally, oxygen is a smaller atom than nitrogen, so the metal-O bond is also shorter and stronger due to the stronger overlap of atomic orbitals.

Using DFT calculations to estimate stability of GaON in aqueous solution is an interesting problem, however it is quite time consuming and requires huge computing resources, as we previously demonstrated for a simpler system in the work entitled “Electronic structure of aqueous two-dimensional photocatalyst” [*npj Comp. Mater.* 7 (1), 47 (2021)]. For the system in this work, we think such an analysis warrants a future study. Finally, the experimental results presented in this work have indeed demonstrated the stability of GaON in aqueous solution.

3. Table S4 : what is the precision of the values of formation energy?

Reply: We thank the reviewer for this question. The values of formation energy calculated in this manuscript are by the optB86b-vdW functional [*Phys. Rev. B* 83 (19), 195131 (2011)]. The optB86b-vdW is a modified exchange-correlation functional of the B88, which is a generalized gradient approximation (GGA) functional. The optB86b-vdW functional was developed by adding a non-local correlation term to the B88 functional to account for van der Waals interactions. The optB86b-vdW functional has been shown to provide improved accuracy for a range of properties, including binding energies, reaction energies, and solid-state properties. It

is widely used in materials science and computational chemistry. On the structure side, the lattice parameter, *i.e.*, a , of bulk GaN is calculated to be 3.198 Å, which is in excellent agreement with the experimental value, 3.189 Å [*Appl. Phys. Lett.* **15** (10), 327-329 (1969)]. On the formation energy side, the optB86b-vdW is limited by the inherent GGA approximations whose numerical uncertainty is roughly 0.05 eV. This uncertainty will not change the sign of the formation energy, which is important for our conclusions of the stability.

4. Authors used k-mesh 13x9x1 for primitive GaN cell m-plane. Was convergence of DOS investigated with respect to k-mesh quality?

Reply: Yes, we indeed checked the k-mesh convergence. The convergence of k points mesh density was tested on the bulk-GaN, as plotted below in **Figure A3**, for four different k points mesh densities, *i.e.*, 9, 11, 13, and 15, with lattice length 3.198 Å. The results suggest a k-point mesh density of 13, or even 9, is sufficient to obtain a converged DOS. Regarding the primitive GaN cell *m*-plane, the lattice length is 3.198 Å, 5.216 Å and 35.000 Å, respectively. Therefore, k-mesh 13×9×1 for primitive GaN cell *m*-plane is large enough to get the convergent DOS.

In fact, a k-point mesh density of 8 is employed for GaN system in the literature, for example, the work entitled “Hybrid functional investigations of band gaps and band alignments for AlN, GaN, InN, and InGaN” [*J. Chem. Phys.* **134** (8), 084703 (2011)] done by the Chris G. Van de Walle group.

Figure A3. Convergence test of k-point mesh density on bulk-GaN. 9,11,13,15 are the four different k points mesh density with the lattice length is 3.198 Å.

5. Reader would acknowledge if the 1.39eV shift in the conduction band would be marked on Fig. 5(b). Was band diagram in Fig.5e sketched or calculated, based on which assumptions?

Reply: Thanks, in the revision, we have marked the 1.39 eV shift in the conduction band in Fig. 5b in the main text. The band diagram in Fig.5e was indeed sketched based on the following considerations:

- 1) The straight lines were sketched according to the calculated band edges of the bulk atoms in the GaN m-plane slab model, which are aligned with the vacuum level.
- 2) The downward band bending was sketched based on the revealed experimental and computational upshift of the fermi level relative to the conduction band when the Ga-oxynitrides is observed on the GaN nanowire. The calculated spatial structure of charge densities of Ga-oxynitrides (Fig. 5d) shows that they are surface states. Therefore, the energy level of the emerging surface states is lower than that of the bulk states of GaN, rendering a downward band bending.

In summary, we wish to thank the reviewers again for helping us strengthen the paper. With the above explanations and revisions, we trust that all the comments/questions are adequately addressed.